

# Arrangement of nearby minima and saddles in the mixed spherical energy landscapes

Jaron Kent-Dobias

Istituto Nazionale di Fisica Nucleare, Sezione di Roma I, Italy

## Abstract

The mixed spherical models were recently found to violate long-held assumptions about mean-field glassy dynamics. In particular, the threshold energy, where most stationary points are marginal and that in the simpler pure models attracts long-time dynamics, seems to lose significance. Here, we compute the typical distribution of stationary points relative to each other in mixed models with a replica symmetric complexity. We examine the stability of nearby points, accounting for the presence of an isolated eigenvalue in their spectrum due to their proximity. Despite finding rich structure not present in the pure models, we find nothing that distinguishes the points that do attract the dynamics. Instead, we find new geometric significance of the old threshold energy, and invalidate pictures of the arrangement of most marginal inherent states into a continuous manifold.

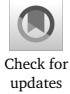

## 1   Introduction

Many systems exhibit "glassiness," characterized by rapid slowing of dynamics over a short parameter interval. These include actual (structural) glasses, spin glasses, certain inference and optimization problems, and more [1–4]. Glassiness is qualitatively understood to arise from structure of an energy or cost landscape, whether due to the proliferation of metastable states, or to the raising of barriers which cause effective dynamic constraints [5–7]. However, in most models there is no known quantitative correspondence between these landscape properties and the dynamic behavior they are purported to describe.

There is such a correspondence in one of the simplest mean-field model of glasses: in the pure spherical models, the dynamic transition corresponds with the energy level at which thermodynamic states attached to marginal inherent states[1] dominate the free energy [8]. At that level, called the *threshold energy* $E_{th}$, slices of the landscape at fixed energy undergo a percolation transition. In fact, this threshold energy is significant in other ways: it attracts the long-time dynamics after quenches in temperature to below the dynamical transition from any starting temperature [9,10]. All of this can be understood in terms of the landscape structure, and namely in the statistics of stationary points of the energy.

In slightly less simple models, the mixed spherical models, the story changes. In these models there are a range of energies with exponentially many marginal minima. It was believed that the energy level at which these marginal minima are the most common type of stationary point would play the same role as the threshold energy in the pure models (in fact we will refer to this energy level as the threshold energy in the mixed models). However, recent work has shown that this is incorrect. Quenches from different starting temperatures above the dynamical transition temperature result in dynamics that approach marginal minima at different energy levels, and the purported threshold does not attract the long-time dynamics in most cases [11,12].

This paper studies the two-point structure of stationary points in the mixed spherical models, or their arrangement relative to each other, previously studied only for the pure models [13]. This gives various kinds of information. When one point is a minimum, we see what other kinds of minima are nearby, and the height of the saddle points that separate them. When both points are saddles, we see the arrangement of barriers relative to each other.

More specifically, one *reference* point is fixed with certain properties. Then, we compute the logarithm of the number of other points constrained to lie at a fixed overlap from the reference point. Constraining the count to points of a fixed overlap from the reference point produces constrained points with atypical properties. For instance, when the required overlap is made sufficiently large, typical constrained points tend to have an isolated eigenvalue pulled out of their spectrum, and its associated eigenvector is correlated with the direction of the reference point. Without the proximity constraint, such an isolated eigenvalue amounts to a large deviation from the spectrum of typical stationary points.

---

[1]For this paper, which focuses on minima, we will take *state* to mean *minimum* or equivalently *inherent state* and not a thermodynamic state. Any discussion of thermodynamic or equilibrium states will explicitly specify this.

In order to address the open problem of what energies attract the long-time dynamics, we focus on the neighborhoods of the marginal minima, to see if there is anything interesting to differentiate sets of them from each other. Though we find rich structure in this population, their properties pivot around the debunked threshold energy, and the apparent attractors of long-time dynamics are not distinguished. Moreover, we show that the usual picture of a marginal 'manifold' of inherent states separated by subextensive barriers [14] is only true at the threshold energy, while at other energies typical marginal minima are far apart and separated by extensive barriers. Therefore, with respect to the problem of dynamics this paper merely deepens the outstanding issues.

In §2 we define the mixed spherical models and outline some of their important properties. In the following section §3, we go over the main results of this work and their interpretation. In §4 we outline the calculation of the two-point complexity and its expansion in the near-neighborhood of a reference point. Details of the calculation of the complexity are in Appendix A. In §5 we introduce a method for calculating the value of an isolated eigenvalue in the spectrum at stationary points, and outline the calculation for the mixed spherical models. Details of this calculation are in Appendix B. Finally, we draw some conclusions about our results in §6. For the interested reader, a comparison between the two-point complexity and the Franz–Parisi potential in the mixed spherical models is presented in Appendix C.

## 2 The model

The mixed spherical models are defined by the Hamiltonian

$$H(\mathbf{s}) = -\sum_p \frac{1}{p!} \sum_{i_1 \cdots i_p}^N J^{(p)}_{i_1 \cdots i_p} s_{i_1} \cdots s_{i_p} \,, \tag{1}$$

where the vectors $\mathbf{s} \in \mathbb{R}^N$ are confined to the sphere $\|\mathbf{s}\|^2 = N$ [15–17]. The coupling coefficients $J$ are fully-connected and random, with zero mean and variance $\overline{(J^{(p)})^2} = a_p p!/2N^{p-1}$ scaled so that the energy is typically extensive. The overbar denotes an average over the coefficients $J$. The factors $a_p$ in the variances are freely chosen constants that define the particular model. For instance, the 'pure' $p$-spin model has $a_{p'} = \delta_{p'p}$. This class of models encompasses all statistically isotropic Gaussian random Hamiltonians defined on the hypersphere.

The covariance between the energy at two different points is a function of the overlap, or dot product, between those points, or

$$\overline{H(\mathbf{s}_1)H(\mathbf{s}_2)} = N f\left(\frac{\mathbf{s}_1 \cdot \mathbf{s}_2}{N}\right), \tag{2}$$

where the function $f$ is defined from the coefficients $a_p$ by

$$f(q) = \frac{1}{2} \sum_p a_p q^p \,. \tag{3}$$

The choice of $f$ has significant effect on the form of equilibrium order in the model, and likewise influences the geometry of stationary points [17, 18].

To enforce the spherical constraint at stationary points, we make use of a Lagrange multiplier $\omega$. This results in the extremal problem

$$H(\mathbf{s}) + \frac{\omega}{2}(\|\mathbf{s}\|^2 - N) \,. \tag{4}$$

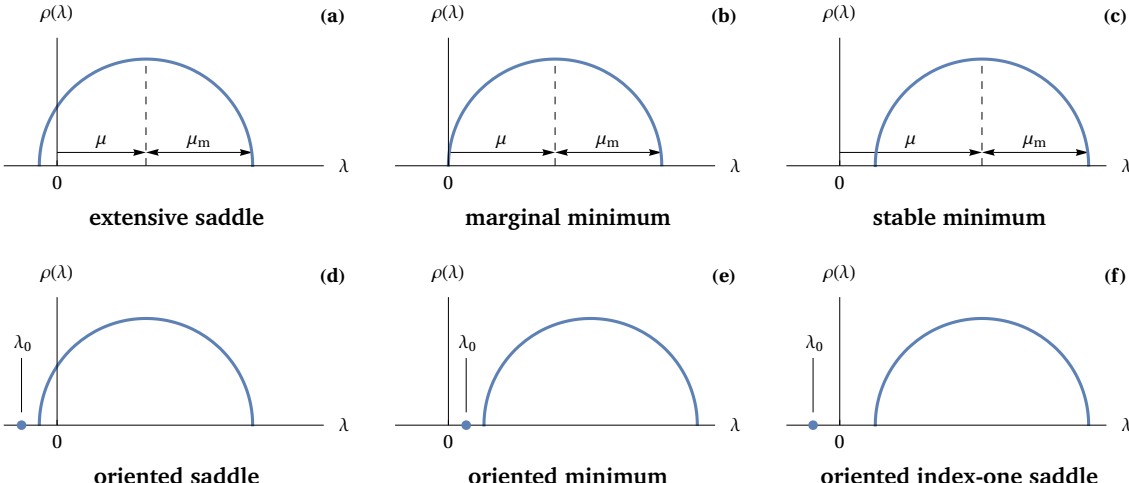

Figure 1: Illustration of the interpretation of the stability $\mu$, which sets the location of the center of the eigenvalue spectrum. In the top row we have spectra without an isolated eigenvalue. **(a)** $\mu < \mu_\mathrm{m}$, there are an extensive number of downward directions, and the associated point is an *extensive saddle*. **(b)** $\mu = \mu_\mathrm{m}$ and we have a *marginal minimum* with asymptotically flat directions. **(c)** $\mu > \mu_\mathrm{m}$, all eigenvalues are positive, and the point is a *stable minimum*. On the bottom we show what happens in the presence of an isolated eigenvalue. **(d)** One eigenvalue leaves the bulk spectrum of a saddle point and it remains a saddle point, but now with an eigenvector correlated with the orientation of the reference vector, so we call this an *oriented saddle*. **(e)** The same happens for a minimum, and we can call it an *oriented minimum*. **(f)** One eigenvalue outside a positive bulk spectrum is negative, destabilizing what would otherwise have been a stable minimum, producing an *oriented index-one saddle*.

The gradient and Hessian at a stationary point are then

$$\nabla H(\mathbf{s}, \omega) = \partial H(\mathbf{s}) + \omega \mathbf{s}, \qquad \mathrm{Hess}\, H(\mathbf{s}, \omega) = \partial\partial H(\mathbf{s}) + \omega I, \qquad (5)$$

where $\partial = \frac{\partial}{\partial \mathbf{s}}$ denotes the derivative with respect to $\mathbf{s}$.

When we count stationary points, we classify them by certain properties. One of these is the energy density $E = H/N$. We will also fix the *stability* $\mu = \frac{1}{N} \mathrm{Tr}\, \mathrm{Hess}\, H$, also known as the radial reaction. In the mixed spherical models, all stationary points have a semicircle law for the eigenvalue spectrum of their Hessians, each with the same width $\mu_\mathrm{m}$, but whose center is shifted by different amounts. Fixing the stability $\mu$ fixes this shift, and therefore fixes the spectrum of the associated stationary point. When the stability is smaller than the width of the spectrum, or $\mu < \mu_\mathrm{m}$, there are an extensive number of negative eigenvalues, and the stationary point is a saddle with a large index whose value is set by the stability. When the stability is greater than the width of the spectrum, or $\mu > \mu_\mathrm{m}$, the semicircle distribution lies only over positive eigenvalues, and unless an isolated eigenvalue leaves the semicircle and becomes negative, the stationary point is a minimum. Finally, when $\mu = \mu_\mathrm{m}$, the edge of the semicircle touches zero and we have marginal minima. Fig. 1 shows what different values of the stability imply about the spectrum at stationary points.

In the pure spherical models, $E$ and $\mu$ cannot be fixed separately: fixing one uniquely fixes the other. This property leads to the great simplification of these models: marginal minima exist *only* at one energy level, and therefore only that energy has the possibility of trapping the long-time dynamics. In generic mixed models this is not the case and at a given energy

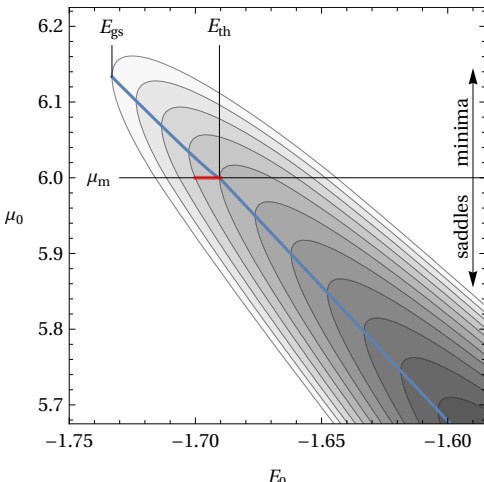

Figure 2: Plot of the complexity (logarithm of the number of stationary points) for the $3 + 4$ mixed spherical model studied in this paper. Energies and stabilities of interest are marked, including the ground state energy $E_{gs}$, the marginal stability $\mu_m$, and the threshold energy $E_{th}$. The blue line shows the location of the most common type of stationary point at each energy level. The highlighted red region shows the approximate range of minima that attract aging dynamics from a quench to zero temperature found in [11].

level $E$ there are many stabilities for which exponentially many marginal points are found. We define the threshold energy $E_{th}$ as the energy at which most stationary points are marginal.[2]

In this study, our numeric examples are drawn exclusively from the model studied in [11], whose covariance function is given by

$$f_{3+4}(q) = \frac{1}{2}\left(q^3 + q^4\right). \tag{6}$$

First, the ordering of its stationary points is like that of the pure spherical models, without any clustering [19]. Second, properties of its long-time dynamics have been extensively studied and are available for comparison. Though the numeric examples all come from the $3+4$ model, the results apply to any model sharing its simple order. The annealed one-point complexity of these models was calculated in [20], and for this model the annealed calculation is expected to be correct.

The one-point complexity of the $3 + 4$ model as a function of energy $E_0$ and stability $\mu_0$ is plotted in Fig. 2. The same plot for a pure $p$-spin model would consist of only a line, because $E_0$ and $\mu_0$ cannot be varied independently. Several important features of the complexity are highlighted: the energies of the ground state $E_{gs}$ and the threshold $E_{th}$, along with the line of marginal stability $\mu_m$. Along the line of marginal stability, energies that attract aging dynamics from different temperatures are highlighted in red. One might expect some feature to mark the ends of this range, something that would differentiate marginal minima that support aging dynamics from those that do not. As indicated in the introduction, the two-point complexity we study in this paper does not produce such a result.

---

[2]Note that crucially this is *not* the energy that has the most marginal stationary points: this energy level with the largest number of marginal points has even more saddles of extensive index. So $E_{th}$ contains a *minority* of the marginal points, even if those marginal points are the *majority* of stationary points with energy $E_{th}$.

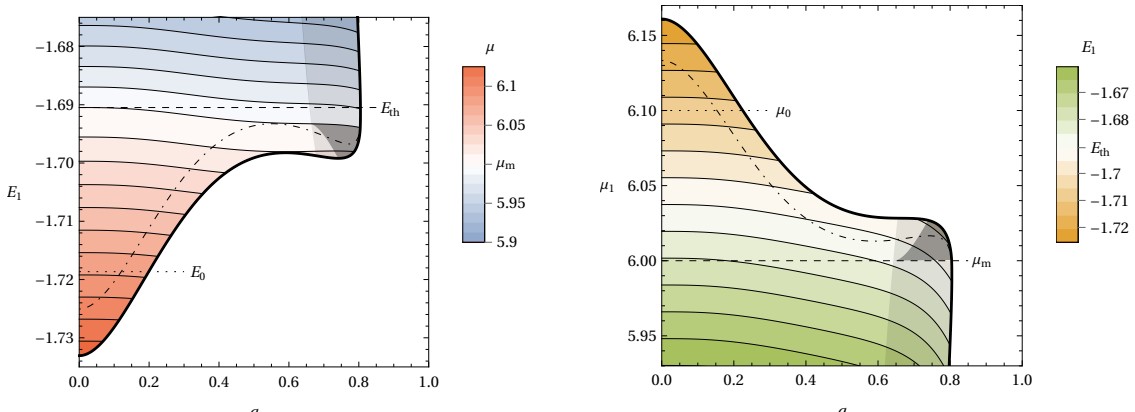

Figure 3: The neighborhood of a reference minimum with $E_0 = -1.71865 < E_{\text{th}}$ and $\mu_0 = 6.1 > \mu_{\text{m}}$. **Left:** The most common type of stationary point lying at fixed overlap $q$ and energy $E_1$ from the reference minimum. The black line gives the smallest or largest energies where neighbors can be found at a given overlap. **Right:** The most common type of stationary point lying at fixed overlap $q$ and stability $\mu_1$ from the reference minimum. Note that this describes a different set of stationary points than shown in the left plot. On both plots, the shading of the righthand part depicts the state of an isolated eigenvalue in the spectrum of the Hessian of the neighboring points. Those more lightly shaded are points with an isolated eigenvalue that does not change their stability, e.g., corresponding with Fig. 1(d-e). The more darkly shaded are oriented index-one saddles, e.g., corresponding with Fig. 1(f). The dot-dashed line on the left plot depicts the trajectory of the solid line on the right plot, and the dot-dashed line on the right plot depicts the trajectory of the solid line on the left plot. In this case, the points lying nearest to the reference minimum are saddles with $\mu < \mu_{\text{m}}$, but with energies smaller than the threshold energy, which makes them an atypical population of saddles.

## 3 Results

Our results stem from the two-point complexity $\Sigma_{12}$, which is defined as the logarithm of the number of stationary points with energy $E_1$ and stability $\mu_1$ that lie at an overlap $q$ with a typical reference stationary point whose energy is $E_0$ and stability is $\mu_0$. When the complexity is positive, there are exponentially many stationary points with the given properties conditioned on the existence of the reference point. When it is zero, there are only order-one such points, and when it is negative there are exponentially few (effectively, none). In the examples below, the boundary of zero complexity between exponentially many and few points is often highlighted, with parameter regions that have negative complexity having no color. Finally, as a result of the condition that the counted points lie with a given proximity to the reference point, their spectrum can be modified by the presence of an isolated eigenvalue, which can change their stability as shown in Fig. 1.

### 3.1 Barriers around deep states

If the reference configuration is a stable minimum, then there is a gap in the overlap between it and its nearest neighbors in configuration space. We can characterize these neighbors as a function of their overlap and stability, with one example seen in Fig. 3. For stable minima, the qualitative results for the pure $p$-spin model continue to hold, with some small modifications [13].

The largest difference between the pure and mixed models is the decoupling of nearby stable points from nearby low-energy points: in the pure $p$-spin model, the left and right panels of Fig. 3 would be identical up to a constant factor $-p$. Instead, for mixed models they differ substantially, as evidenced by the dot-dashed lines in both plots that in the pure models would correspond exactly with the solid lines. One significant consequence of this difference is the diminished significance of the threshold energy $E_{\text{th}}$: in the left panel, marginal minima of the threshold energy are the most common among unconstrained points with $q = 0$, but marginal minima of lower energy are more common in the vicinity of the example reference minimum. In the pure models, all marginal minima are at the threshold energy.

The nearest neighbor points are always oriented saddles, sometimes saddles with an extensive index and sometimes index-one saddles (Fig. 1(d, f)). This is a result of the persistent presence of a negative isolated eigenvalue in the spectrum of the nearest neighbors, e.g., as in the shaded regions of Fig. 3. Like in the pure models, the minimum energy and maximum stability of nearby points are not monotonic in $q$: there is a range of overlap where the minimum energy of neighbors decreases with overlap. The transition from stable minima to index-one saddles along the line of lowest-energy states occurs at its local minimum, another similarity with the pure models [13]. This point is interesting because it describes the properties of the nearest stable minima to the reference point. It is not clear why the local minimum of the boundary coincides with this point or what implications that has for behavior.

## 3.2 Grouping of saddle points

At stabilities lower than the marginal stability one finds saddles with an extensive index. Though, being unstable, saddles are not attractors of dynamics, their properties do influence out-of-equilibrium dynamics. For example, high-index saddle points are stationed at the boundaries between different basins of attraction of gradient flow, and for a given basin the flow between adjacent saddle points defines a complex with implications for the landscape topology [21].

Other stationary points are found at arbitrarily small distances from a reference extensive saddle point, with a linear pseudogap in their complexity. The energy and stability of these near neighbors approach that of the reference point as the difference in overlap $\Delta q$ is brought to zero. However, the approach of the energy and stability are at different rates: the energy difference between the reference and its neighbors shrinks like $\Delta q^2$, while the stability difference shrinks like $\Delta q$. This means that the near neighborhood of saddle points is dominated by the presence of other saddle points at very similar energy, but relatively variable index. Descending between saddles must simultaneously lower the index and the energy, but if the energy and stability change with the same order of magnitude, the connected saddle points must lie at a macroscopic distance from each other. This makes it impossible to use the properties of nearest neighbors to draw inferences about the way saddle points are connected by gradient flow.

## 3.3 Geometry of marginal states

The set of marginal states is of special interest. First, marginal states are known to attract physical and algorithmic dynamics [22]. Second, they have more structure than in the pure models, with different types of marginal states being found at different energies. We find, surprisingly, that the properties of marginal states pivot around the threshold energy, the energy at which most stationary points are marginal, but which is not significant for aging dynamics.

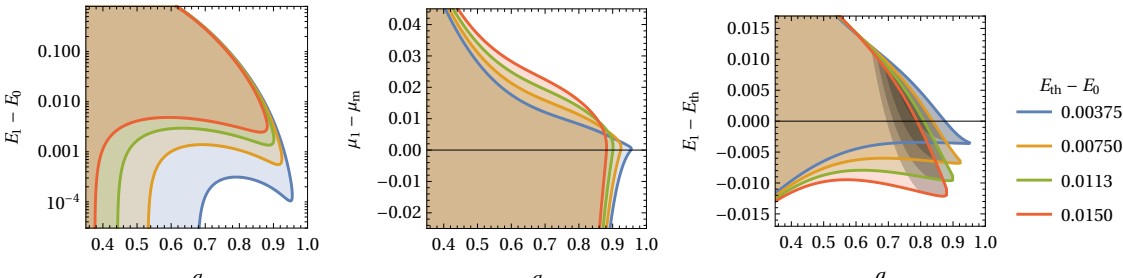

Figure 4: The neighborhood of marginal states at several energies below the threshold energy. **Left:** The range of energies $E_1$ at which nearby states are found. For any $E_0 < E_{\text{th}}$, the nearest class of states is at an extensive distance, and their energies are higher than that of the reference configuration. **Center:** The range of stabilities $\mu_1$ at which nearby states are found. For $E_0$ near the threshold, the nearest states are always index-one saddles with $\mu > \mu_{\text{m}}$, but as the overlap gap widens their population becomes model-dependent. **Right:** The range of energies at which *other* marginal states are found. Here, the more darkly shaded regions denote where an isolated eigenvalue appears. Marginal states below the threshold are always separated by a gap in their overlap.

- **Energies below the threshold.** These marginal states have a macroscopic gap in their overlap with nearby minima and saddles. Their nearest-neighbor stationary points are saddles with an oriented direction, and their nearest neighbors always have a higher energy density than the reference state. Fig. 4 shows examples of the neighborhoods of these marginal minima.

- **Energies above the threshold.** These marginal states have neighboring stationary points at arbitrarily close distance, with a quadratic pseudogap in their complexity. Their nearest neighbors are *strictly* saddle points with an extensive number of downward directions and their nearest neighbors always have a higher energy density than the reference state. The nearest neighboring marginal states have an overlap gap with the reference state. Fig. 5 shows examples of the neighborhoods of these marginal minima.

- **At the threshold energy.** These marginal states have neighboring stationary points at arbitrarily close distance, with a cubic pseudogap in their complexity. The nearest ones include oriented saddle points with an extensive number of downward directions, and oriented stable and marginal minima. Though most of the nearest states are found at higher energies, they can also be found at the same energy density as the reference state. Fig. 6 shows examples of the neighborhoods of these marginal states.

This leads us to some general conclusions. First, at all energy densities except at the threshold energy, *typical marginal minima are separated by extensive energy barriers*. Therefore, the picture of a marginal *manifold* of many (even all) marginal states lying arbitrarily close and being connected by subextensive energy barriers can only describe the collection of marginal minima at the threshold energy, which is an atypical population of marginal minima. At energies both below and above the threshold energy, typical marginal minima are isolated from each other.[3]

---

[3]We must put a small caveat here: for any combination of energy and stability of the reference point, this calculation admits order-one other marginal minima to lie a subextensive distance from the reference point. For such a population of points, $\Sigma_{12} = 0$ and $q = 1$, which is always a permitted solution when at least one marginal direction exists. These points are separated by small barriers from one another, but they also cover a vanishing

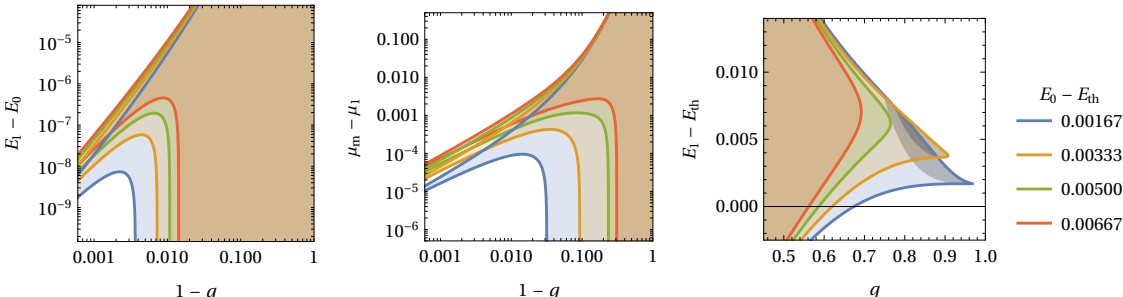

Figure 5: The neighborhood of marginal states at several energies above the threshold energy. **Left:** The range of energies $E_1$ at which nearby states are found. For any $E_0 > E_{\text{th}}$, there always exists a $q$ sufficiently close to one such that the nearby states have strictly greater energy than the reference state. **Center:** The range of stabilities $\mu_1$ at which nearby states are found. There is always a sufficiently large overlap beyond which all nearby states are saddle with an extensive number of downward directions. **Right:** The range of energies at which *other* marginal states are found. Here, the more darkly shaded regions denote where an isolated eigenvalue appears. Marginal states above the threshold are always separated by a gap in their overlap.

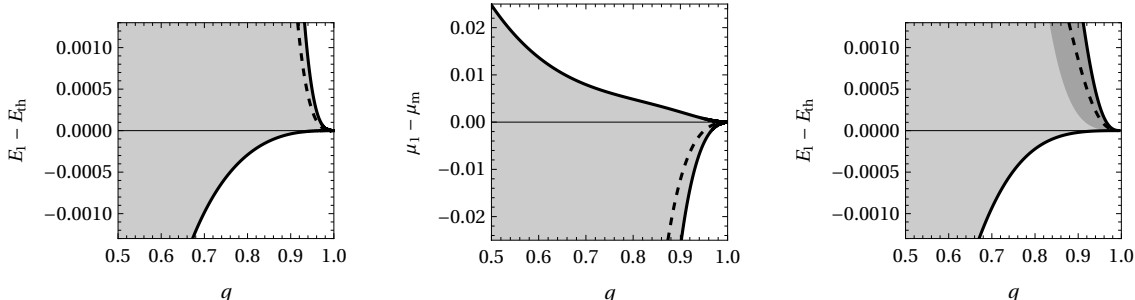

Figure 6: The neighborhood of marginal minima at the threshold energy $E_0 = E_{\text{th}}$. In all plots, the dashed lines show the population of most common neighbors at the given overlap $q$. **Left:** The range of energies $E_1$ at which nearby points are found. The approach of both the minimum and maximum energies goes like $(1-q)^3$. **Center:** The range of stabilities $\mu_1$ at which nearby points are found. The approach of both limits goes like $(1-q)^2$. **Right:** The range of nearby marginal minima. The more darkly shaded region denotes where an isolated eigenvalue appears. Marginal minima at the threshold lie asymptotically close together.

This has implications for how quench dynamics should be interpreted. When typical marginal states are approached above the threshold energy, they must have been via the neighborhood of saddles with an extensive index, not other marginal states. On the other hand, typical marginal states approached below the threshold energy must be reached after an extensive distance in configuration space without encountering any stationary point. The geometric conditions of the neighborhoods above and below are quite different, but the observed aging dynamics don't appear to qualitatively change [11, 12]. Therefore, if the marginal minima attracting dynamics are typical ones, the conditions in the neighborhood of the marginal minimum eventually reached at infinite time appear to be irrelevant for the nature of aging dynamics at any finite time.

---

piece of configuration space, and each such cluster of points is isolated by extensive barriers from each other cluster in the way described above. To move on a 'manifold' of nearby marginal minima within such a cluster cannot describe aging, since the overlap with the initial condition will never change from one.

A version of this story was told a long time ago by the authors of [14], who write on aging in the pure spherical models where the limit of $N \to \infty$ is taken before that of $t \to \infty$: "it is important to remark that this [...] does *not* mean that the system relaxes into a near-threshold state: at all finite times an infinite system has a Hessian with an *infinite* number of directions in which the energy is a maximum. [...] We have seen that the saddles separating threshold minima are typically $O(N^{1/3})$ above the threshold level, while the energy is at all finite times $O(N)$ above this level." In the present case of the mixed spherical models, where [11] has shown aging dynamics asymptotically approaching marginal states that we have shown have $O(N)$ saddles separating them, this lesson must be taken all the more seriously.

On the other hand, it is possible that *atypical* marginal minima are relevant for attracting the dynamics. Studying these points would require a different kind of computation, where the fixed reference point is abandoned and both points are treated on equal footing. Such a calculation is beyond the scope of this paper, but is clear fodder for future research.

## 4 Calculation of the two-point complexity

To calculate the two-point complexity, we extend a common method for counting stationary points: the Kac–Rice method [23, 24]. The basic idea is that stationary points of a function can be counted by integrating a Dirac $\delta$-function containing the function's gradient over its domain. Because the argument of the $\delta$-function is nonlinear in the integration variable, it must be weighted by the determinant of its Jacobian, which happens to be the Hessian of the function. It is not common that this procedure can be analytically carried out for an explicit function. However, in the spherical models it can be carried out *on average*.

In order to lighten notation, we introduce the Kac–Rice measure

$$d\nu_H(\mathbf{s}, \omega) = 2\, d\mathbf{s}\, d\omega\, \delta(\|\mathbf{s}\|^2 - N)\, \delta\big(\nabla H(\mathbf{s}, \omega)\big) \big| \det \operatorname{Hess} H(\mathbf{s}, \omega) \big|, \tag{7}$$

containing the $\delta$-function of the gradient and determinant of the Hessian of the Hamiltonian, along with a $\delta$-function enforcing the spherical constraint. If integrated over configuration space, $\mathcal{N}_H = \int d\nu_H(\mathbf{s}, \omega)$ gives the total number of stationary points in the function. The Kac–Rice method has been used by in many studies to analyze the geometry of random functions [25–28]. More interesting is the measure conditioned on the energy density $E$ and stability $\mu$ of the points,

$$d\nu_H(\mathbf{s}, \omega \mid E, \mu) = d\nu_H(\mathbf{s}, \omega)\, \delta\big(NE - H(\mathbf{s})\big)\, \delta\big(N\mu - \operatorname{Tr} \operatorname{Hess} H(\mathbf{s}, \omega)\big). \tag{8}$$

While $\mu$ is strictly the trace of the Hessian, we call it the stability because in this family of models all stationary points have a bulk spectrum of the same shape, shifted by different constants. The stability $\mu$ sets this shift, and therefore determines the stiffness of minima and the typical index of saddle points. See Fig. 1 for examples.

We want the typical number of stationary points with energy density $E_1$ and stability $\mu_1$ that lie a fixed overlap $q$ from a reference stationary point of energy density $E_0$ and stability $\mu_0$. For a *typical* number, we cannot average the total number $\mathcal{N}_H$, which is exponentially large in $N$ and therefore can be biased by atypical examples. Therefore, we will average the logarithm of this number. The two-point complexity is therefore defined by

$$\Sigma_{12} = \frac{1}{N} \overline{\int \frac{d\nu_H(\boldsymbol{\sigma}, \varsigma \mid E_0, \mu_0)}{\int d\nu_H(\boldsymbol{\sigma}', \varsigma' \mid E_0, \mu_0)} \log\left( \int d\nu_H(\mathbf{s}, \omega \mid E_1, \mu_1)\, \delta(Nq - \boldsymbol{\sigma} \cdot \mathbf{s}) \right)}. \tag{9}$$

Inside the logarithm, the measure (8) is integrated with the further condition that $\mathbf{s}$ has a certain overlap with the reference configuration $\boldsymbol{\sigma}$. The entire expression is then integrated

over $\boldsymbol{\sigma}$ again by the Kac–Rice measure, then divided by a normalization. This is equivalent to summing the logarithm over all stationary points $\boldsymbol{\sigma}$ with the given properties, then dividing by their total number, i.e., an average.

It is difficult to take the disorder average of anything that is not an exponential integral. The normalization integral over $\boldsymbol{\sigma}'$ in the denominator and the integral inside the logarithm both pose a problem. Each can be treated using the replica trick: $\lim_{m\to 0} x^{m-1} = \frac{1}{x}$ and $\lim_{n\to 0}\frac{\partial}{\partial n} x^n = \log x$. Applying these transformations, we have

$$\Sigma_{12} = \frac{1}{N}\lim_{n\to 0}\lim_{m\to 0}\frac{\partial}{\partial n}\overline{\int\left(\prod_{b=1}^{m} d\,\nu_H(\boldsymbol{\sigma}_b,\varsigma_b \mid E_0,\mu_0)\right)\left(\prod_{a=1}^{n} d\,\nu_H(\mathbf{s}_a,\omega_a \mid E_1,\mu_1)\,\delta(Nq-\boldsymbol{\sigma}_1\cdot\mathbf{s}_a)\right)}.$$
(10)

Note that among the $\boldsymbol{\sigma}$ replicas, $\boldsymbol{\sigma}_1$ is special. The $m-1$ replicas $\boldsymbol{\sigma}_2,\ldots,\boldsymbol{\sigma}_m$ correspond to the replicated normalization integral over $\boldsymbol{\sigma}'$, which is completely uncoupled from $\mathbf{s}$. The variable $\boldsymbol{\sigma}_1$ is not a replica: it is the same as $\boldsymbol{\sigma}$ in (9), and is the only of the $\boldsymbol{\sigma}$s that couples with $\mathbf{s}$.

This expression can now be averaged over the disordered couplings, and its integration evaluated using the saddle point method. We must assume the form of order among the replicas $\mathbf{s}$ and $\boldsymbol{\sigma}$, and we take them to be replica symmetric. Replica symmetry means that at the saddle point, all distinct pairs of replicas have the same overlap. This choice is well-motivated for the $3+4$ and similar models. Details of the calculation can be found in Appendix A.

The resulting expression for the complexity, which must still be extremized over the order parameters $\hat{\beta}_1$, $r^{01}$, $r_{\mathrm{d}}^{11}$, $r_0^{11}$, and $q_0^{11}$, is

$$\Sigma_{12}(E_0,\mu_0,E_1,\mu_1,q) = \operatorname*{extremum}_{\hat{\beta}_1,r_{\mathrm{d}}^{11},r_0^{11},r^{01},q_0^{11}}\left\{\mathcal{D}(\mu_1)-\frac{1}{2}+\hat{\beta}_1 E_1 - r_{\mathrm{d}}^{11}\mu_1 + \hat{\beta}_1\big(r_{\mathrm{d}}^{11}f'(1)-r_0^{11}f'(q_0^{11})\big)\right.$$

$$+\hat{\beta}_0\hat{\beta}_1 f(q)+(\hat{\beta}_0 r^{01}+\hat{\beta}_1 r^{10}+r_{\mathrm{d}}^{00}r^{01})f'(q)+\frac{r_{\mathrm{d}}^{11}-r_0^{11}}{1-q_0^{11}}(r^{10}-qr_{\mathrm{d}}^{00})f'(q)$$

$$+\frac{1}{2}\left[\hat{\beta}_1^2\big(f(1)-f(q_0^{11})\big)+(r_{\mathrm{d}}^{11})^2 f''(1)+2r^{01}r^{10}f''(q)-(r_0^{11})^2 f''(q_0^{11})+\frac{(r^{10}-qr_{\mathrm{d}}^{00})^2}{1-q_0^{11}}f'(1)\right.$$

$$+\frac{1-q^2}{1-q_0^{11}}+\left((r^{01})^2-\frac{r_{\mathrm{d}}^{11}-r_0^{11}}{1-q_0^{11}}\left(2qr^{01}-\frac{(1-q^2)r_0^{11}-(q_0^{11}-q^2)r_{\mathrm{d}}^{11}}{1-q_0^{11}}\right)\right)\big(f'(1)-f'(q_{22}^{(0)})\big)$$

$$\left.\left.-\frac{1}{f'(1)}\frac{f'(1)^2-f'(q)^2}{f'(1)-f'(q_0^{11})}+\frac{r_{\mathrm{d}}^{11}-r_0^{11}}{1-q_0^{11}}\big(r_{\mathrm{d}}^{11}f'(1)-r_0^{11}f'(q_0^{11})\big)+\log\left(\frac{1-q_{11}^0}{f'(1)-f'(q_{11}^0)}\right)\right]\right\},$$
(11)

where the function $\mathcal{D}$ is defined in (A.4) of Appendix A. It is possible to further extremize this expression over all the other variables but $q_0^{11}$, for which the saddle point conditions have a unique solution. However, the resulting expression is quite complicated and provides no insight. In fact, the numeric root-finding problem is more stable when preserving these parameters, rather than analytically eliminating them.

In practice, the complexity can be calculated in two ways. First, the extremal problem can be done numerically, initializing from $q=0$ where the problem reduces to that of the single-point complexity of points with energy $E_1$ and stability $\mu_1$, which has an analytical solution. Then small steps in $q$ or other parameters are taken to analytically continue the solution. This is how the data in all the plots of this paper was produced. Second, the complexity can be calculated in the near neighborhood of a reference point by expanding in powers of small $\Delta q = 1-q$. This expansion indicates when nearby points can be found at arbitrarily small distance, and in that case gives the form of the pseudogap in their complexity.

If there is no overlap gap between the reference point and its nearest neighbors, their complexity can be calculated by an expansion in $1-q$. First, we'll use this method to describe

the most common type of stationary point in the close vicinity of a reference point. These are given by further maximizing the two-point complexity over the energy $E_1$ and stability $\mu_1$ of the nearby points. This gives the conditions

$$\hat{\beta}_1 = 0, \qquad\qquad \mu_1 = 2r_{\mathrm{d}}^{11}f''(1), \qquad\qquad (12)$$

where the second condition is only true for $\mu_1^2 \leq \mu_{\mathrm{m}}^2$, i.e., when the nearby points are saddle points or marginal minima. When these conditions are inserted into the complexity, an expansion is made in small $1-q$, and the saddle point in the remaining parameters is taken, the result is

$$\Sigma_{12} = \frac{f'''(1)}{8f''(1)^2}(\mu_{\mathrm{m}}^2 - \mu_0^2)\left(\sqrt{2 + \frac{2f''(1)\big(f''(1)-f'(1)\big)}{f'''(1)f'(1)}} - 1\right)(1-q) + O\big((1-q)^2\big), \quad (13)$$

independent of $E_0$. Notice that slope of the complexity is positive for $\mu_0 < \mu_{\mathrm{m}}$ and vanishes when the stability of the reference point approaches the marginal stability. This implies that extensive saddle points have arbitrarily close neighbors with a linear pseudogap, while stable minima have an overlap gap with their nearest neighbors. For marginal minima, the existence of arbitrarily close neighbors must be decided at quadratic order and higher.

To describe the properties of these most common neighbors, it is convenient to first make a definition. The population of stationary points that are most common at each energy (the blue line in Fig. 2) have the relation

$$E_{\mathrm{dom}}(\mu_0) = -\frac{f'(1)^2 + f(1)\big(f''(1)-f'(1)\big)}{2f''(1)f'(1)}\mu_0, \qquad\qquad (14)$$

between $E_0$ and $\mu_0$ for $\mu_0^2 \leq \mu_{\mathrm{m}}^2$. Using this definition, the energy and stability of the most common neighbors at small $\Delta q$ are

$$E_1 = E_0 + \frac{1}{2}\frac{v_f}{u_f}\big(E_0 - E_{\mathrm{dom}}(\mu_0)\big)(1-q)^2 + O\big((1-q)^3\big), \qquad\qquad (15)$$

$$\mu_1 = \mu_0 - \frac{v_f}{u_f}\big(E_0 - E_{\mathrm{dom}}(\mu_0)\big)(1-q) + O\big((1-q)^2\big), \qquad\qquad (16)$$

where $v_f$ and $u_f$ are positive functionals of $f$ defined in (A.21) of Appendix A. The most common neighboring saddles to a reference saddle are much nearer to the reference in energy ($\Delta q^2$) than in stability ($\Delta q$). In fact, this scaling also holds for all neighbors of a reference saddle, not just the most common.

Because both expressions are proportional to $E_0 - E_{\mathrm{dom}}(\mu_0)$, whether the energy and stability of nearby points increases or decreases from that of the reference point depends only on whether the energy of the reference point is above or below that of the most common population at the same stability, i.e., to the right or left of the blue line in Fig. 2. In particular, since $E_{\mathrm{dom}}(\mu_{\mathrm{m}}) = E_{\mathrm{th}}$, the threshold energy is also the pivot around which the points asymptotically nearby marginal minima change their properties.

To examine better the population of marginal points, it is necessary to look at the next term in the series of the complexity with $\Delta q$, since the linear coefficient becomes zero at the marginal line. When $\mu = \mu_{\mathrm{m}}$, the quadratic term in the expansion for the dominant population of near neighbors is

$$\Sigma_{12} = \frac{1}{2}\frac{f'''(1)v_f}{f''(1)^{3/2}u_f}\left(\sqrt{\frac{2\big[f'(1)(f'''(1)-f''(1)) + f''(1)^2\big]}{f'(1)f'''(1)}} - 1\right)(E_0 - E_{\mathrm{th}})(1-q)^2 + O\big((1-q)^3\big).$$
$$(17)$$

This coefficient is positive when $E > E_{\text{th}}$ and negative when $E < E_{\text{th}}$. Therefore, marginal minima whose energy $E_0$ is greater than the threshold have neighbors at arbitrarily close distance with a quadratic pseudogap, while those whose energy is less than the threshold have an overlap gap. Exactly at the threshold the cubic term in the expansion is necessary; it is not insightful to share explicitly but it is positive for the $3 + 4$ and similar models.

The properties of the nearby states above the threshold can be further quantified. Though we know from (15) and (16) that the most common nearby points at small distance are extensive saddle points with higher energy than the reference point, we do not know what other kinds of stationary points might also be found in close proximity. Could these marginal minima sit at very small distance from other marginal minima? The answer is that the very near neighbors are exclusively extensive saddles of higher energy. Therefore, the marginal minima with energies above the threshold energy also have overlap gaps with one another. These results on the range of possible neighbors are elaborated in Appendix A.5.

## 5  Finding the isolated eigenvalue

The two-point complexity $\Sigma_{12}$ depends on the spectrum at both stationary points through the determinant of their Hessians, but only on the bulk of the distribution. This bulk is unaffected by the conditions of energy and proximity. However, these conditions give rise to small-rank perturbations to the Hessian, which can cause a subextensive number of eigenvalues to leave the bulk. We study the possibility of *one* stray eigenvalue.

We use a technique recently developed to find the smallest eigenvalue of random matrices [29]. One defines an artificial quadratic statistical mechanics model with configurations defined on the sphere, whose interaction tensor is given by the matrix of interest. By construction, the ground state is located in the direction of the eigenvector associated with the smallest eigenvalue, and the ground state energy is proportional to that eigenvalue.

Our matrix of interest is the Hessian evaluated at a stationary point of the mixed spherical model, conditioned on the relative position, energies, and stabilities discussed above. We must restrict the artificial spherical model to lie in the tangent plane of the 'real' spherical configuration space at the point of interest, to avoid our eigenvector pointing in a direction that violates the spherical constraint. A sketch of the setup is shown in Fig. 7. The free energy of the artificial model given a point $\mathbf{s}$ and for a specific realization of the disordered Hamiltonian is

$$
\begin{aligned}
\beta F_H(\beta \mid \mathbf{s}, \omega) &= -\frac{1}{N} \log\left( \int d\mathbf{x}\, \delta(\mathbf{x} \cdot \mathbf{s}) \delta(\|\mathbf{x}\|^2 - N) \exp\left\{ -\beta \frac{1}{2} \mathbf{x}^T \operatorname{Hess} H(\mathbf{s}, \omega) \mathbf{x} \right\} \right) \\
&= -\lim_{\ell \to 0} \frac{1}{N} \frac{\partial}{\partial \ell} \int \left[ \prod_{\alpha=1}^{\ell} d\mathbf{x}_\alpha\, \delta(\mathbf{x}_\alpha^T \mathbf{s}) \delta(N - \mathbf{x}_\alpha^T \mathbf{x}_\alpha) \exp\left\{ -\beta \frac{1}{2} \mathbf{x}_\alpha^T \big( \partial\partial H(\mathbf{s}) + \omega I \big) \mathbf{x}_\alpha \right\} \right],
\end{aligned}
$$
(18)

where the first $\delta$-function keeps the configurations in the tangent plane, and the second enforces the spherical constraint. We have anticipated treating the logarithm with replicas. We are interested in points $\mathbf{s}$ that have certain properties: they are stationary points of $H$ with given energy density and stability, and fixed overlap from a reference configuration $\boldsymbol{\sigma}$. We therefore average the free energy above over such points, giving

$$
\begin{aligned}
F_H(\beta \mid E_1, \mu_1, q, \boldsymbol{\sigma}) &= \int \frac{d\nu_H(\mathbf{s}, \omega \mid E_1, \mu_1) \delta(Nq - \boldsymbol{\sigma} \cdot \mathbf{s})}{\int d\nu_H(\mathbf{s}', \omega' \mid E_1, \mu_1) \delta(Nq - \boldsymbol{\sigma} \cdot \mathbf{s}')} F_H(\beta \mid \mathbf{s}, \omega) \\
&= \lim_{n \to 0} \int \left[ \prod_{a=1}^{n} d\nu_H(\mathbf{s}_a, \omega_a \mid E_1, \mu_1) \delta(Nq - \boldsymbol{\sigma} \cdot \mathbf{s}_a) \right] F_H(\beta \mid \mathbf{s}_1, \omega_1),
\end{aligned}
$$
(19)

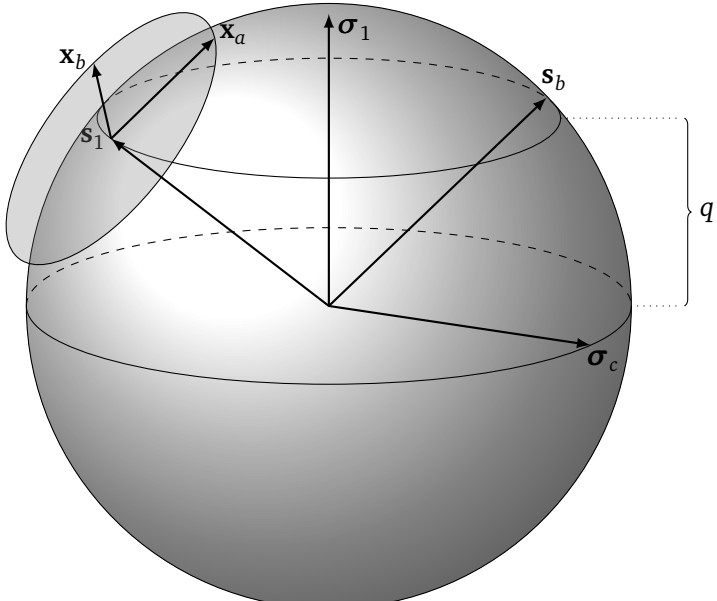

Figure 7: A sketch of the vectors involved in the calculation of the isolated eigenvalue. All replicas $\mathbf{x}$, which correspond with candidate eigenvectors of the Hessian evaluated at $\mathbf{s}_1$, sit in an $N-2$ sphere corresponding with the tangent plane (not to scale) of the first $\mathbf{s}$ replica. All of the $\mathbf{s}$ replicas lie on the sphere, constrained to be at fixed overlap $q$ with the first of the $\boldsymbol{\sigma}$ replicas, the reference configuration. All of the $\boldsymbol{\sigma}$ replicas lie on the sphere.

again anticipating the use of replicas. Finally, the reference configuration $\boldsymbol{\sigma}$ should itself be a stationary point of $H$ with its own energy density and stability, as before. Averaging over these conditions gives

$$
\begin{aligned}
F_H(\beta \mid E_0, \mu_0, E_1, \mu_1, q) &= \int \frac{d\nu_H(\boldsymbol{\sigma}, \varsigma \mid E_0, \mu_0)}{\int d\nu_H(\boldsymbol{\sigma}', \varsigma' \mid E_0, \mu_0)} F_H(\beta \mid E_1, \mu_1, q, \boldsymbol{\sigma}) \\
&= \lim_{m \to 0} \int \left[ \prod_{a=1}^{m} d\nu_H(\boldsymbol{\sigma}_a, \varsigma_a \mid E_0, \mu_0) \right] F_H(\beta \mid E_1, \mu_1, q, \boldsymbol{\sigma}_1).
\end{aligned}
\tag{20}
$$

This formidable expression is now ready to be averaged over the disordered Hamiltonians $H$. Once averaged, the minimum eigenvalue of the conditioned Hessian is then given by twice the ground state energy, or

$$
\lambda_{\min} = 2 \lim_{\beta \to \infty} \overline{F_H(\beta \mid E_0, \mu_0, E_1, \mu_1, q)}.
\tag{21}
$$

For this calculation, there are three different sets of replicated variables. Note that, as for the computation of the complexity, the $\boldsymbol{\sigma}_1$ and $\mathbf{s}_1$ replicas are *special*. The first again is the only of the $\boldsymbol{\sigma}$ replicas constrained to lie at fixed overlap with *all* the $\mathbf{s}$ replicas, and the second is the only of the $\mathbf{s}$ replicas at which the Hessian is evaluated.

The calculation of this minimum eigenvalue is very similar to that of the complexity. The details of this calculation can be found in Appendix B. The result for the minimum eigenvalue is given by

$$
\lambda_{\min} = \mu_1 - \left( y + \frac{1}{y} f''(1) \right),
\tag{22}
$$

where $y$ is an order parameter whose value is set by the saddle-point conditions

$$
0 = -f''(1) + y^2(1 - \mathcal{X}^T C \mathcal{X}), \qquad\qquad 0 = (B - yC)\mathcal{X},
\tag{23}
$$

for $\mathcal{X} \in \mathbb{R}^5$ a vector of order parameters, and $B$ and $C$ are $5 \times 5$ matrices whose elements are explicit functions of the order parameters from the two-point complexity problem and of $f$ and its derivatives. The matrices $B$ and $C$ are given in (B.13) and (B.14) of Appendix B.

There is a trivial solution for $\mathcal{X} = 0$ and $y^2 = f''(1)$. This results in a minimum eigenvalue

$$\lambda_{\min} = \mu_1 - \sqrt{4f''(1)} = \mu_1 - \mu_{\mathrm{m}}, \tag{24}$$

that corresponds with the bottom edge of the semicircle distribution. This is the correct solution in the absence of an isolated eigenvalue. Any solution corresponding to the presence of an isolated eigenvalue must have nonzero $\mathcal{X}$. The only way to satisfy this with the second of the saddle conditions (23) is for $y$ such that one of the eigenvalues of $B - yC$ is zero. Under these circumstances, if the normalized eigenvector associated with the zero eigenvector is $\hat{\mathcal{X}}_0$, then $\mathcal{X} = \|\mathcal{X}_0\| \hat{\mathcal{X}}_0$ is a solution. The magnitude $\|\mathcal{X}_0\|$ of this solution is set by the first saddle point condition, namely

$$\|\mathcal{X}_0\|^2 = \frac{1}{\hat{\mathcal{X}}_0^T C \hat{\mathcal{X}}_0} \left(1 - \frac{f''(1)}{y^2}\right). \tag{25}$$

In practice, we find that $\hat{\mathcal{X}}_0^T C \hat{\mathcal{X}}_0$ is positive at the saddle point. Therefore, for the solution to exist we must have $y^2 \geq f''(1)$. In practice, there is at most one $y$ which produces a zero eigenvalue of $B - yC$ and satisfies this inequality, so the solution seems to be unique.

With this solution, we simultaneously find the smallest eigenvalue and information about the orientation of its associated eigenvector: namely, its overlap $q_{\min}$ with the tangent vector that points directly from one stationary point to the other. This information is encoded the order parameter vector $\mathcal{X}$, and the details of how it is computed can be found at the end of Appendix B. The emergence of an isolated eigenvalue and its associated eigenvector are shown in Fig. 8, for the same reference point properties that were used in Fig. 3. For small overlaps, the minimum eigenvalue corresponds with bottom of the semicircle distribution, or the trivial solution. As the overlap is increased, one eigenvalue continuously leaves the spectrum, with an eigenvector whose overlap with the vector between stationary points also grows continuously from zero.

Though the two-point complexity $\Sigma_{12}$ fails to distinguish the marginal minima at the limits of aging dynamics, one might imagine that something related to the isolated eigenvalue might succeed in distinguishing them. This does not appear to be the case. Above and below the threshold energy, the nature of the isolated eigenvalue of nearest neighbors does not change: it is always present and varies continuously. There is an energy both above and below the threshold where the nearest marginal states transition from having an isolated eigenvalue to not having one; see for instance in the right panel of Fig. 5 that the grey region vanishes. One might reason that this could change the connectivity of nearby marginal-like states and thereby the aging dynamics. However, the energies where these changes occur are not close to the limits of aging dynamics measured by [11], so that reasoning is wrong.

## 6 Conclusion

We have computed the complexity of neighboring stationary points for the mixed spherical models. When we studied the neighborhoods of marginal minima, we found something striking: only those at the threshold energy have other marginal minima nearby. For the many marginal minima away from the threshold (including the exponential majority), there is a gap in overlap between them.

This has implications for pictures of relaxation and aging. In most $p + s$ models studied, quenches from infinite to zero temperature (gradient descent starting from a random point)

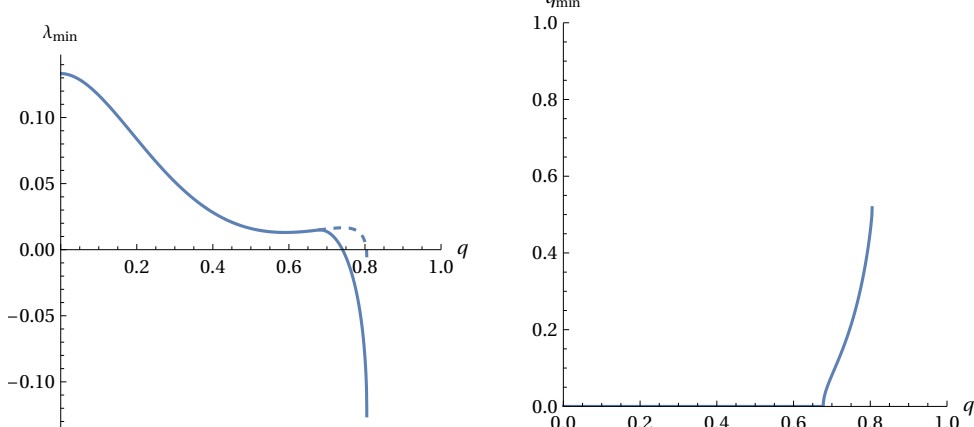

Figure 8: Properties of the isolated eigenvalue and the overlap of its associated eigenvector with the direction of the reference point. These curves correspond with the lower solid curve in Fig. 3. **Left:** The value of the minimum eigenvalue as a function of overlap. The dashed line shows the continuation of the bottom of the semicircle. Where the dashed line separates from the solid line, the isolated eigenvalue has appeared. **Right:** The overlap between the eigenvector associated with the minimum eigenvalue and the direction of the reference point. The overlap is zero until an isolated eigenvalue appears, and then it grows continuously until the nearest neighbor is reached.

relax towards marginal states with energies above the threshold energy [22], while at least in some models a quench to zero temperature from a temperature around the dynamic transition relaxes towards marginal states with energies below the threshold energy [11,12]. We found (see especially Figs. 4 and 5) that the neighborhoods of marginal states above and below the threshold are quite different, and yet the emergent aging behavior relaxing toward states above and below the threshold seem to be the same. Therefore, aging dynamics appears to be insensitive to the neighborhood of the marginal state being approached. To understand something better about why certain states attract the dynamics in certain situations, nonlocal information, like the structure of their entire basin of attraction, seems vital.

It is possible that replica symmetry breaking among the constrained stationary points could change the details of the two-point complexity of very nearby states. Indeed, it is difficult to rule out RSB in complexity calculations. However, such corrections would not change the overarching conclusions of this paper, namely that most marginal minima are separated from each other by a macroscopic overlap gap and high barriers. This is because the replica symmetric complexity bounds any RSB complexities from above, and so RSB corrections can only decrease the complexity. Therefore, the overlap gaps, which correspond to regions of negative complexity, cannot be removed by a more detailed saddle point Ansatz.

Our calculation studied the neighborhood of typical reference points with the given energy and stability. However, it is possible that marginal minima with atypical neighborhoods actually attract the dynamics, as has been argued in certain neural networks [30, 31]. To determine this, a different type of calculation is needed. As our calculation is akin to the quenched Franz–Parisi potential, study of atypical neighborhoods would entail something like the annealed Franz–Parisi approach, i.e.,

$$\Sigma^*(E_0, \mu_0, E_1, \mu_1, q) = \frac{1}{N} \overline{\log \left( \int d\nu_H(\boldsymbol{\sigma}, \varsigma \,|\, E_0, \mu_0)\, d\nu_H(\mathbf{s}, \omega \,|\, E_1, \mu_1)\, \delta(Nq - \boldsymbol{\sigma} \cdot \mathbf{s}) \right)}, \quad (26)$$

which puts the two points on equal footing. This calculation and exploration of the atypical neighborhoods it reveals is a clear future direction.

The methods developed in this paper are straightforwardly (if not easily) generalized to landscapes with replica symmetry broken complexities [28]. We suspect that many of the qualitative features of this study would persist, with neighboring states being divided into different clusters based on the RSB order but with the basic presence or absence of overlap gaps and the nature of the stability of near-neighbors remaining unchanged. Interesting structure might emerge in the arrangement of marginal states in FRSB systems, where the ground state itself is marginal and coincides with the threshold.

## Acknowledgments

The author would like to thank Valentina Ros, Giampaolo Folena, Chiara Cammarota, and Jorge Kurchan for useful discussions related to this work.

**Funding information** JK-D is supported by a DYNSYSMATH Specific Initiative by the INFN.

## A  Details of the calculation for the two-point complexity

The two-point complexity defined in (9) consists of the average over integrals containing of products of Dirac $\delta$-functions and determinants of Hessians. To compute it, we first split the factors into two groups: one group that contains any dependence on the Hessian (the determinants and the $\delta$-functions fixing the stabilities) and a second group containing all other $\delta$-functions. The average over disorder for the two groups of factors can be made independently, which is described in subsections A.1 and A.2 for the Hessian and other factors, respectively.

Once the average is made over disorder, the result is an exponential integral that depends only on scalar products between the replicated configurations $\mathbf{s}$ and $\boldsymbol{\sigma}$ and their conjugate fields. The explicit dependence on these microscopic configurations is removed using a Hubbard–Stratonovich transformation, which replaces the scalar products with overlap order parameters. This is described in subsection A.3. Finally, the complexity is an exponential integral over several order parameter fields, and is amenable to evaluation by a saddle point method, detailed in subsection A.4.

### A.1  The Hessian factors

The factors dependant on the Hessian can be averaged over disorder using results from random matrix theory. The double partial derivatives of the energy are Gaussian with the variance

$$\overline{(\partial_i \partial_j H(\mathbf{s}))^2} = \frac{1}{N} f''(1), \tag{A.1}$$

which means that the matrix of partial derivatives belongs to the GOE class. Its spectrum is given by the Wigner semicircle

$$\rho(\lambda) = \begin{cases} \frac{2}{\pi} \sqrt{1 - \left(\frac{\lambda}{\mu_{\mathrm{m}}}\right)^2}, & \lambda^2 \le \mu_{\mathrm{m}}^2, \\ 0, & \text{otherwise}, \end{cases} \tag{A.2}$$

with radius $\mu_{\mathrm{m}} = \sqrt{4f''(1)}$. Since the Hessian differs from the matrix of partial derivatives by adding the constant diagonal matrix $\omega I$, it follows that the spectrum of the Hessian is a Wigner semicircle shifted by $\omega$, or $\rho(\lambda + \omega)$.

The average over factors depending on the Hessian alone can be made separately from those depending on the gradient or energy, since for random Gaussian fields the Hessian is

independent of these [27]. In principle the fact that we have conditioned the Hessian to belong to stationary points of certain energy, stability, and proximity to another stationary point will modify its statistics, but these changes will only appear at subleading order in $N$ [13]. This is because the conditioning amounts to a rank-one perturbation to the Hessian matrix, which does not affect the bulk of its spectrum. At leading order, the expectations related to different replicas factorize, each yielding

$$\overline{\left|\det\operatorname{Hess}H(\mathbf{s},\omega)\right|\delta\left(N\mu-\operatorname{Tr}\operatorname{Hess}H(\mathbf{s},\omega)\right)}=e^{N\int d\lambda\,\rho(\lambda+\mu)\log|\lambda|}\delta(N\mu-N\omega),\qquad(\text{A.3})$$

Therefore, each of the Lagrange multipliers is fixed to one of the stabilities $\mu$. We define the function

$$
\begin{aligned}
\mathcal{D}(\mu)&=\int d\lambda\,\rho(\lambda+\mu)\log|\lambda|\\
&=\begin{cases}
\frac{1}{2}+\log\left(\frac{1}{2}\mu_{\mathrm{m}}\right)+\frac{\mu^2}{\mu_{\mathrm{m}}^2}, & \mu^2\le\mu_{\mathrm{m}}^2,\\
\frac{1}{2}+\log\left(\frac{1}{2}\mu_{\mathrm{m}}\right)+\frac{\mu^2}{\mu_{\mathrm{m}}^2}-\left|\frac{\mu}{\mu_{\mathrm{m}}}\right|\sqrt{\left(\frac{\mu}{\mu_{\mathrm{m}}}\right)^2-1}-\log\left(\left|\frac{\mu}{\mu_{\mathrm{m}}}\right|-\sqrt{\left(\frac{\mu}{\mu_{\mathrm{m}}}\right)^2-1}\right), & \mu^2>\mu_{\mathrm{m}}^2,
\end{cases}
\end{aligned}
$$
$$(\text{A.4})$$

and using it the full factor due to the Hessians can be written

$$e^{Nm\mathcal{D}(\mu_0)+Nn\mathcal{D}(\mu_1)}\left[\prod_a^m\delta(N\mu_0-N\varsigma_a)\right]\left[\prod_a^n\delta(N\mu_1-N\omega_a)\right].\qquad(\text{A.5})$$

## A.2 The other factors

The other factors consist of $\delta$-functions of the gradient and $\delta$-functions containing the energy and spherical constraints. We take advantage of the Fourier representation of the $\delta$-function to express each of them as an exponential integral over an auxiliary field. For instance,

$$\delta\left(\nabla H(\mathbf{s},\mu_1)\right)=\int\frac{d\hat{\mathbf{s}}}{(2\pi)^N}e^{i\hat{\mathbf{s}}\cdot\nabla H(\mathbf{s},\mu_1)},\qquad(\text{A.6})$$

replaces a $\delta$-function of the gradient by introducing the auxiliary field $\hat{\mathbf{s}}$. Carrying out such a transformation to each of the remaining factors gives an exponential integrand of the form

$$e^{Nm\hat{\beta}_0E_0+Nn\hat{\beta}_1E_1-\sum_a^m\left[(\boldsymbol{\sigma}_a\cdot\hat{\boldsymbol{\sigma}}_a)\mu_0-\frac{1}{2}\hat{\mu}_0(N-\boldsymbol{\sigma}_a\cdot\boldsymbol{\sigma}_a)\right]-\sum_a^n\left[(\mathbf{s}_a\cdot\hat{\mathbf{s}}_a)\mu_1-\frac{1}{2}\hat{\mu}_1(N-\mathbf{s}_a\cdot\mathbf{s}_a)-\frac{1}{2}\hat{\mu}_{12}(Nq-\boldsymbol{\sigma}_1\cdot\mathbf{s}_a)\right]+\int d\mathbf{t}\,\mathcal{O}(\mathbf{t})H(\mathbf{t})},$$
$$(\text{A.7})$$

where we have introduced the linear operator

$$\mathcal{O}(\mathbf{t})=\sum_a^m\delta(\mathbf{t}-\boldsymbol{\sigma}_a)\left(i\hat{\boldsymbol{\sigma}}_a\cdot\partial_{\mathbf{t}}-\hat{\beta}_0\right)+\sum_a^n\delta(\mathbf{t}-\mathbf{s}_a)\left(i\hat{\mathbf{s}}_a\cdot\partial_{\mathbf{t}}-\hat{\beta}_1\right),\qquad(\text{A.8})$$

consolidating all of the $H$-dependent terms. Here the $\hat{\beta}$s are the fields auxiliary to the energy constraints, the $\hat{\mu}$s are auxiliary to the spherical and overlap constraints, and the $\hat{\boldsymbol{\sigma}}$s and $\hat{\mathbf{s}}$s are auxiliary to the constraints that the gradient be zero. We have written the $H$-dependent terms in this strange form for the ease of taking the average over $H$: since it is Gaussian-correlated, it follows that

$$\overline{e^{\int d\mathbf{t}\,\mathcal{O}(\mathbf{t})H(\mathbf{t})}}=e^{\frac{1}{2}\int d\mathbf{t}\,d\mathbf{t}'\,\mathcal{O}(\mathbf{t})\mathcal{O}(\mathbf{t}')\overline{H(\mathbf{t})H(\mathbf{t}')}}=e^{N\frac{1}{2}\int d\mathbf{t}\,d\mathbf{t}'\,\mathcal{O}(\mathbf{t})\mathcal{O}(\mathbf{t}')f\left(\frac{\mathbf{t}\cdot\mathbf{t}'}{N}\right)}.\qquad(\text{A.9})$$

It remains only to apply the doubled operators to $f$ and then evaluate the simple integrals over the $\delta$ measures. We do not include these details, which were carried out with computer algebra software. The result of this calculation is found in the effective action (A.15), where it contributes all terms besides the functions $\mathcal{D}$ contributed by the Hessian terms in the previous section and the logarithms contributed by the Hubbard–Stratonovich transformation of the next section.

## A.3   Hubbard–Stratonovich

Having expanded the resulting expression, we are left with an argument in the exponential which is a function of scalar products between the fields $\mathbf{s}$, $\hat{\mathbf{s}}$, $\boldsymbol{\sigma}$, and $\hat{\boldsymbol{\sigma}}$. We will change integration coordinates from these fields to matrix fields given by their scalar products, defined as

$$
\begin{aligned}
C^{00}_{ab} &= \frac{1}{N}\boldsymbol{\sigma}_a \cdot \boldsymbol{\sigma}_b, & R^{00}_{ab} &= -i\frac{1}{N}\boldsymbol{\sigma}_a \cdot \hat{\boldsymbol{\sigma}}_b, & D^{00}_{ab} &= \frac{1}{N}\hat{\boldsymbol{\sigma}}_a \cdot \hat{\boldsymbol{\sigma}}_b, & \\
C^{01}_{ab} &= \frac{1}{N}\boldsymbol{\sigma}_a \cdot \mathbf{s}_b, & R^{01}_{ab} &= -i\frac{1}{N}\boldsymbol{\sigma}_a \cdot \hat{\mathbf{s}}_b, & R^{10}_{ab} &= -i\frac{1}{N}\hat{\boldsymbol{\sigma}}_a \cdot \mathbf{s}_b, & D^{01}_{ab} &= \frac{1}{N}\hat{\boldsymbol{\sigma}}_a \cdot \hat{\mathbf{s}}_b, \\
C^{11}_{ab} &= \frac{1}{N}\mathbf{s}_a \cdot \mathbf{s}_b, & R^{11}_{ab} &= -i\frac{1}{N}\mathbf{s}_a \cdot \hat{\mathbf{s}}_b, & D^{11}_{ab} &= \frac{1}{N}\hat{\mathbf{s}}_a \cdot \hat{\mathbf{s}}_b.
\end{aligned} \tag{A.10}
$$

We insert into the integral the product of $\delta$-functions enforcing these definitions, integrated over the new matrix fields, which is equivalent to multiplying by one. For example, one such factor of one is given by

$$
1 = \int dC^{00} \frac{1}{N^{m^2}} \prod_{ab}^{m} \delta(N C^{00}_{ab} - \boldsymbol{\sigma}_a \cdot \boldsymbol{\sigma}_b). \tag{A.11}
$$

Once this is done, the many scalar products appearing throughout the integrand can be replaced by the matrix fields. The only dependence of the original vector fields is from these new $\delta$-functions. These are treated schematically in following way: let $\{\mathbf{a}_a\} = \{\mathbf{s}_a, \boldsymbol{\sigma}_a, \hat{\mathbf{s}}_a, \hat{\boldsymbol{\sigma}}_a\}$ index all of the original vector fields, and let $Q_{ab} = \frac{1}{N}\mathbf{a}_a \cdot \mathbf{a}_b$ likewise concatenate all of the matrix fields. Then the $\delta$-functions described above can be promoted to an exponential integral of the form

$$
\int d\mathbf{a}\, d\hat{Q}\, e^{N\frac{1}{2}\operatorname{Tr}\hat{Q}Q - \frac{1}{2}\mathbf{a}^T \hat{Q}\mathbf{a}}, \tag{A.12}
$$

using an auxiliary matrix field $\hat{Q}$. The integral over the vector fields $\mathbf{a}$ is Gaussian and can be evaluated, giving

$$
\int d\hat{Q}\, e^{N\operatorname{Tr}\hat{Q}Q}(\det\hat{Q})^{-N/2} = \int d\hat{Q}\, e^{\frac{1}{2}N(\operatorname{Tr}\hat{Q}Q - \log\det\hat{Q})}. \tag{A.13}
$$

Finally, the integral over $\hat{Q}$ can be evaluated using the saddle point method, giving $\hat{Q} = Q^{-1}$. Therefore, the term contributed to the effective action as a result of the transformation is

$$
\frac{1}{2}\log\det Q = \frac{1}{2}\log\det
\begin{bmatrix}
C^{00} & iR^{00} & C^{01} & iR^{01} \\
iR^{00} & D^{00} & iR^{10} & D^{01} \\
C^{01} & iR^{10} & C^{11} & iR^{11} \\
iR^{01} & D^{01} & iR^{11} & D^{11}
\end{bmatrix}. \tag{A.14}
$$

## A.4   Replica Ansatz and saddle point

After the transformation of the previous section, the complexity has been brought to the form of an exponential integral over the matrix order parameters (A.10), proportional to $N$. We are therefore in the position to evaluate this integral using a saddle point method. We will always assume that the square matrices $C^{00}$, $R^{00}$, $D^{00}$, $C^{11}$, $R^{11}$, and $D^{11}$ are hierarchical matrices, i.e., of the Parisi form, with each set of three sharing the same structure. In particular, we immediately define $c_d^{00}$, $r_d^{00}$, $d_d^{00}$, $c_d^{11}$, $r_d^{11}$, and $d_d^{11}$ as the value of the diagonal elements of these matrices, respectively. Note that $c_d^{00} = c_d^{11} = 1$ due to the spherical constraint.

In this paper, we focus on models with a replica symmetric complexity, but many of the intermediate formulae are valid for arbitrary replica symmetry breakings. At most 1RSB in equilibrium is guaranteed if the function $\chi(q) = f''(q)^{-1/2}$ is convex [16]. The complexity at the ground state must reflect the structure of equilibrium, and therefore be replica symmetric. Recent work has found that the complexity of saddle points can have other RSB orders even when the ground state is replica symmetric, but the $3+4$ model has a safely replica symmetric complexity everywhere [19].

Defining the 'block' fields $\mathcal{Q}_{00} = (\hat{\beta}_0, \hat{\mu}_0, C^{00}, R^{00}, D^{00})$, $\mathcal{Q}_{11} = (\hat{\beta}_1, \hat{\mu}_1, C^{11}, R^{11}, D^{11})$, and $\mathcal{Q}_{01} = (\hat{\mu}_{01}, C^{01}, R^{01}, R^{10}, D^{01})$ the resulting complexity is

$$\Sigma_{12} = \frac{1}{N} \lim_{n\to 0} \lim_{m\to 0} \frac{\partial}{\partial n} \int d\mathcal{Q}_{00}\, d\mathcal{Q}_{11}\, d\mathcal{Q}_{01}\, e^{Nm\mathcal{S}_0(\mathcal{Q}_{00}) + Nn\mathcal{S}_1(\mathcal{Q}_{11}, \mathcal{Q}_{01}|\mathcal{Q}_{00})}, \tag{A.15}$$

where

$$\begin{aligned}
\mathcal{S}_0(\mathcal{Q}_{00}) &= \hat{\beta}_0 E_0 - r_{\mathrm{d}}^{00}\mu_0 - \frac{1}{2}\hat{\mu}_0(1 - c_{\mathrm{d}}^{00}) + \mathcal{D}(\mu_0) \\
&\quad + \frac{1}{m}\left\{\frac{1}{2}\sum_{ab}^m \left[\hat{\beta}_1^2 f(C_{ab}^{00}) + (2\hat{\beta}_1 R_{ab}^{00} - D_{ab}^{00})f'(C_{ab}^{00}) + (R_{ab}^{00})^2 f''(C_{ab}^{00})\right] + \frac{1}{2}\log\det\begin{bmatrix} C^{00} & iR^{00} \\ iR^{00} & D^{00} \end{bmatrix}\right\},
\end{aligned} \tag{A.16}$$

is the action for the ordinary, one-point complexity, and the remainder is given by

$$\begin{aligned}
\mathcal{S}_1(\mathcal{Q}_{11}, \mathcal{Q}_{01} \mid \mathcal{Q}_{00}) &= \hat{\beta}_1 E_1 - r_{\mathrm{d}}^{11}\mu_1 - \frac{1}{2}\hat{\mu}_1(1 - c_{\mathrm{d}}^{11}) + \mathcal{D}(\mu_1) \\
&\quad + \frac{1}{n}\sum_b^n\left\{-\frac{1}{2}\hat{\mu}_{12}(q - C_{1b}^{01})\right. \\
&\qquad\qquad \left. + \sum_a^m \left[\hat{\beta}_0\hat{\beta}_1 f(C_{ab}^{01}) + (\hat{\beta}_0 R_{ab}^{01} + \hat{\beta}_1 R_{ab}^{10} - D_{ab}^{01})f'(C_{ab}^{01}) + R_{ab}^{01}R_{ab}^{10}f''(C_{ab}^{01})\right]\right\} \\
&\quad + \frac{1}{n}\left\{\frac{1}{2}\sum_{ab}^n\left[\hat{\beta}_1^2 f(C_{ab}^{11}) + (2\hat{\beta}_1 R_{ab}^{11} - D_{ab}^{11})f'(C_{ab}^{11}) + (R_{ab}^{11})^2 f''(C_{ab}^{11})\right]\right. \\
&\qquad \left. + \frac{1}{2}\log\det\left(\begin{bmatrix} C^{11} & iR^{11} \\ iR^{11} & D^{11} \end{bmatrix} - \begin{bmatrix} C^{01} & iR^{01} \\ iR^{10} & D^{01} \end{bmatrix}^T \begin{bmatrix} C^{00} & iR^{00} \\ iR^{00} & D^{00} \end{bmatrix}^{-1} \begin{bmatrix} C^{01} & iR^{01} \\ iR^{10} & D^{01} \end{bmatrix}\right)\right\}.
\end{aligned} \tag{A.17}$$

Because of the structure of this problem in the twin limits of $m$ and $n$ to zero, the parameters $\mathcal{Q}_{00}$ can be evaluated at a saddle point of $\mathcal{S}_0$ alone. This means that these parameters will take the same value they take when the ordinary, 1-point complexity is calculated. For a replica symmetric complexity of the reference point, this results in

$$\hat{\beta}_0 = -\frac{\mu_0 f'(1) + E_0\big(f'(1) + f''(1)\big)}{u_f}, \tag{A.18}$$

$$r_{\mathrm{d}}^{00} = \frac{\mu_0 f(1) + E_0 f'(1)}{u_f}, \tag{A.19}$$

$$d_{\mathrm{d}}^{00} = \frac{1}{f'(1)} - \left(\frac{\mu_0 f(1) + E_0 f'(1)}{u_f}\right)^2, \tag{A.20}$$

where we define for brevity (here and elsewhere) the constants

$$u_f = f(1)\big(f'(1) + f''(1)\big) - f'(1)^2, \qquad v_f = f'(1)\big(f''(1) + f'''(1)\big) - f''(1)^2. \tag{A.21}$$

Note that because the coefficients of $f$ must be nonnegative for $f$ to be a sensible covariance, both $u_f$ and $v_f$ are strictly positive.[4]

In general, we except the $m \times n$ matrices $C^{01}$, $R^{01}$, $R^{10}$, and $D^{01}$ to have constant *rows* of length $n$, with blocks of rows corresponding to the RSB structure of the single-point complexity for the model. For the scope of this paper, where we restrict ourselves to replica symmetric complexities, they have the following form at the saddle point:

$$
C^{01} = \begin{bmatrix} q & \cdots & q \\ 0 & \cdots & 0 \\ \vdots & \ddots & \vdots \\ 0 & \cdots & 0 \end{bmatrix} \begin{matrix} \uparrow \\ m-1 \\ \downarrow \end{matrix} \, , \qquad R^{01} = \begin{bmatrix} r_{01} & \cdots & r_{01} \\ 0 & \cdots & 0 \\ \vdots & \ddots & \vdots \\ 0 & \cdots & 0 \end{bmatrix} ,
$$

$$
R^{10} = \begin{bmatrix} r_{10} & \cdots & r_{10} \\ 0 & \cdots & 0 \\ \vdots & \ddots & \vdots \\ 0 & \cdots & 0 \end{bmatrix} , \qquad D^{01} = \begin{bmatrix} d_{01} & \cdots & d_{01} \\ 0 & \cdots & 0 \\ \vdots & \ddots & \vdots \\ 0 & \cdots & 0 \end{bmatrix} ,
$$

(A.22)

where only the first row is nonzero. The other entries, which correspond to the completely uncorrelated replicas in an RSB picture, are all zero because uncorrelated vectors on the sphere are orthogonal.

The most challenging part of inserting our replica symmetric Ansatz is the volume element in the log det, which involves the product and inverse of block replica matrices. The inverse of block hierarchical matrix is still a block hierarchical matrix, since

$$
\begin{bmatrix} C^{00} & iR^{00} \\ iR^{00} & D^{00} \end{bmatrix}^{-1} = \begin{bmatrix} (C^{00}D^{00} + R^{00}R^{00})^{-1}D^{00} & -i(C^{00}D^{00} + R^{00}R^{00})^{-1}R^{00} \\ -i(C^{00}D^{00} + R^{00}R^{00})^{-1}R^{00} & (C^{00}D^{00} + R^{00}R^{00})^{-1}C^{00} \end{bmatrix} , \qquad (A.23)
$$

and hierarchical matrices are closed under inverses and products. Because of the structure of the 01 matrices, the volume element will depend only on the diagonals of the matrices in this inverse block matrix. If we define

$$
\tilde{c}_{\mathrm{d}}^{00} = [(C^{00}D^{00} + R^{00}R^{00})^{-1}C^{00}]_{\mathrm{d}} , \qquad (A.24)
$$

$$
\tilde{r}_{\mathrm{d}}^{00} = [(C^{00}D^{00} + R^{00}R^{00})^{-1}R^{00}]_{\mathrm{d}} , \qquad (A.25)
$$

$$
\tilde{d}_{\mathrm{d}}^{00} = [(C^{00}D^{00} + R^{00}R^{00})^{-1}D^{00}]_{\mathrm{d}} , \qquad (A.26)
$$

as the diagonals of the blocks of the inverse matrix, then the result of the product is

$$
\begin{bmatrix} C^{01} & iR^{01} \\ iR^{10} & D^{01} \end{bmatrix}^T \begin{bmatrix} C^{00} & iR^{00} \\ iR^{00} & D^{00} \end{bmatrix}^{-1} \begin{bmatrix} C^{01} & iR^{01} \\ iR^{10} & D^{01} \end{bmatrix}
$$
$$
= \begin{bmatrix} q^2\tilde{d}_{\mathrm{d}}^{00} + 2qr_{10}\tilde{r}_{\mathrm{d}}^{00} - r_{10}^2\tilde{d}_{\mathrm{d}}^{00} & i\left[d_{01}(r_{10}\tilde{c}_{\mathrm{d}}^{00} - q\tilde{r}_{\mathrm{d}}^{00}) + r_{01}(r_{10}\tilde{r}_{\mathrm{d}}^{00} + q\tilde{d}_{\mathrm{d}}^{00})\right] \\ i\left[d_{01}(r_{10}\tilde{c}_{\mathrm{d}}^{00} - q\tilde{r}_{\mathrm{d}}^{00}) + r_{01}(r_{10}\tilde{r}_{\mathrm{d}}^{00} + q\tilde{d}_{\mathrm{d}}^{00})\right] & d_{01}^2\tilde{c}_{\mathrm{d}}^{00} + 2r_{01}d_{01}\tilde{r}_{\mathrm{d}}^{00} - r_{01}^2\tilde{d}_{\mathrm{d}}^{00} \end{bmatrix} ,
$$
(A.27)

where each block is a constant $n \times n$ matrix. Because the matrices $C^{00}$, $R^{00}$, and $D^{00}$ are diagonal in the replica symmetric case, the diagonals of the blocks above take a simple form:

$$
\tilde{c}_{\mathrm{d}}^{00} = f'(1) , \qquad \tilde{r}_{\mathrm{d}}^{00} = r_{\mathrm{d}}^{00} f'(1) , \qquad \tilde{d}_{\mathrm{d}}^{00} = d_{\mathrm{d}}^{00} f'(1) . \qquad (A.28)
$$

Once these expressions are inserted into the complexity, the limits of $n$ and $m$ to zero can be taken, and the parameters from $D^{01}$ and $D^{11}$ can be extremized explicitly. The result is (11) from section 4 of the main text.

---

[4]Note also that $u_f = v_f = 0$ if $f$ is a homogeneous polynomial as in the pure models. These expressions are invalid for the pure models because $\mu_0$ and $E_0$ cannot be fixed independently; we would have done the equivalent of inserting two identical $\delta$-functions. For the pure models, the terms $\hat{\beta}_0$ and $\hat{\beta}_1$ must be set to zero in our prior formulae (as if the energy was not constrained) and then the saddle point taken.

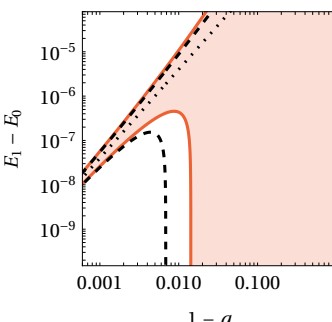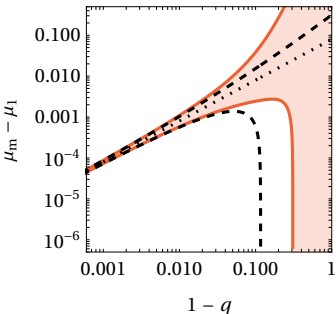

Figure 9: Demonstration of the convergence of the $(1-q)$-expansion for marginal reference minima. Solid lines and shaded region show are the same as in Fig. 5 for $E_0 - E_{\text{th}} \simeq 0.00667$. The dotted lines show the expansion of most common neighbors, while the dashed lines in both plots show the expansion for the minimum and maximum energies and stabilities found at given $q$.

### A.5 The range of energies and stabilities of nearby points

The range of parameters that result in a positive complexity is found by taking the complexity (11) and further requiring that $\Sigma_{12} = 0$. The maximum and minimum stability are then found by maximizing this constrained expression over the energy, while the maximum and minimum energy are found by maximizing it over the stability. In the small-$\Delta q$ expansion outlined in §4, these ranges can be computed analytically. We share the results here for the neighbors to marginal minima with energies greater than the threshold energy, and confirm that the analytically computed ranges match those found numerically.

The limit of stability in which nearby points are found to marginal minima above the threshold are given by $\mu_1 = \mu_{\text{m}} + \delta\mu_1(1-q) \pm \delta\mu_2(1-q)^{3/2} + O\big((1-q)^2\big)$ where $\delta\mu_1$ is given by the coefficient in (16) and

$$\delta\mu_2 = \frac{v_f}{f'(1)f''(1)^{3/4}}\sqrt{\frac{E_0 - E_{\text{th}}}{2}\frac{2f''(1)\big(f''(1)-f'(1)\big)+f'(1)f'''(1)}{u_f}}. \tag{A.29}$$

Since the limits differ from the most common points at higher order in $\Delta q$, nearby points are of the same kind as the most common population. Similarly, one finds that the energy lies in the range $E_1 = E_0 + \delta E_1(1-q)^2 \pm \delta E_2(1-q)^{5/2} + O\big((1-q)^3\big)$ for $\delta E_1$ given by the coefficient in (15) and

$$\delta E_2 = \frac{\sqrt{E_0 - E_{\text{th}}}}{4f'(1)f''(1)^{3/4}}\bigg(\frac{v_f}{3u_f}\big[f'(1)(2f''(1)-(2-(2-\delta q_0)\delta q_0)f'''(1))-2f''(1)^2\big]$$

$$\times \big[f'(1)\big(6f''(1)+(18-(6-\delta q_0)\delta q_0)f'''(1)\big)-6f''(1)^2\big]\bigg)^{\frac{1}{2}}, \tag{A.30}$$

and $\delta q_0$ is the coefficient in the expansion $q_0 = 1 - \delta q_0(1-q) + O((1-q)^2)$ and is given by the real root to the quintic equation

$$0 = ((16-(6-\delta q_0)\delta q_0)\delta q_0 - 12)f'(1)f'''(1) - 2\delta q_0(f''(1)-f'(1))f''(1). \tag{A.31}$$

These predictions from the small $1-q$ expansion are compared with numeric saddle points for the complexity of marginal minima in Fig. 9, and the results agree well at small $1-q$.

# B  Details of calculation for the isolated eigenvalue

Many of the steps in the evaluation of the isolated eigenvalue are similar to those from the evaluation of the two-point complexity: the treatment of the average over disorder and the Hubbard–Stratonovich transformation follow the exact same reasoning. We will not repeat the details of techniques that were already reported in the previous appendix.

The treatment of the factors in the average over disorder proceeds as it does for the complexity in A.2, now with the disorder-dependent terms captured in the linear operator

$$\mathcal{O}(\mathbf{t}) = \sum_a^m \delta(\mathbf{t}-\boldsymbol{\sigma}_a)(i\hat{\boldsymbol{\sigma}}_a \cdot \partial_{\mathbf{t}} - \hat{\beta}_0) + \sum_b^n \delta(\mathbf{t}-\mathbf{s}_b)(i\hat{\mathbf{s}}_b \cdot \partial_{\mathbf{t}} - \hat{\beta}_1) - \frac{1}{2}\delta(\mathbf{t}-\mathbf{s}_1)\beta \sum_c^\ell (\mathbf{x}_c \cdot \partial_{\mathbf{t}})^2 \,, \quad \text{(B.1)}$$

that is applied to $H$ by integrating over $\mathbf{t} \in \mathbb{R}^N$. The resulting expression for the integrand produces dependencies on the scalar products in (A.10) and on the new scalar products involving the tangent plane vectors $\mathbf{x}$:

$$A_{ab} = \frac{1}{N}\mathbf{x}_a \cdot \mathbf{x}_b \,, \quad X_{ab}^0 = \frac{1}{N}\boldsymbol{\sigma}_a \cdot \mathbf{x}_b \,, \quad \hat{X}_{ab}^0 = -i\frac{1}{N}\hat{\boldsymbol{\sigma}}_a \cdot \mathbf{x}_b \,, \quad X_{ab}^1 = \frac{1}{N}\mathbf{s}_a \cdot \mathbf{x}_b \,, \quad \hat{X}_{ab}^1 = -i\frac{1}{N}\hat{\mathbf{s}}_a \cdot \mathbf{x}_b \,.$$
$$\text{(B.2)}$$

Replacing the original variables using a Hubbard–Stratonovich transformation then proceeds like it did for the complexity in subsection A.3. Defining as before a block variable $\mathcal{Q}_x = (A, X^0, \hat{X}^0, X^1, \hat{X}^1)$ and consolidating the previous block variables $\mathcal{Q} = (\mathcal{Q}_{00}, \mathcal{Q}_{01}, \mathcal{Q}_{11})$, we can write the minimum eigenvalue schematically as

$$\lambda_{\min} = -2 \lim_{\beta \to \infty} \lim_{\substack{\ell \to 0 \\ m \to 0 \\ n \to 0}} \frac{\partial}{\partial \ell} \frac{1}{\beta N} \int d\mathcal{Q} \, d\mathcal{Q}_x \, e^{N[m\mathcal{S}_0(\mathcal{Q}_{00}) + n\mathcal{S}_1(\mathcal{Q}_{11},\mathcal{Q}_{01}|\mathcal{Q}_{00}) + \ell\mathcal{S}_x(\mathcal{Q}_x|\mathcal{Q}_{00},\mathcal{Q}_{01},\mathcal{Q}_{11})]} \,, \quad \text{(B.3)}$$

where $\mathcal{S}_0$ is given by (A.16), $\mathcal{S}_1$ is given by (A.17), and the new action $\mathcal{S}_x$ is given by

$$
\begin{aligned}
\ell\mathcal{S}_x(\mathcal{Q}_x \mid \mathcal{Q}) = &-\frac{1}{2}\ell\beta\mu_1 + \frac{1}{2}\beta \sum_b^\ell \left\{ \frac{1}{2}\beta f''(1) \sum_a^l A_{ab}^2 \right. \\
&+ \sum_a^m \left[ \left(\hat{\beta}_0 f''(C_{a1}^{01}) + R_{a1}^{10} f'''(C_{a1}^{01})\right)(X_{ab}^0)^2 + 2f''(C_{a1}^{01})X_{ab}^0 \hat{X}_{ab}^0 \right] \\
&+ \left. \sum_a^n \left[ \left(\hat{\beta}_1 f''(C_{a1}^{11}) + R_{a1}^{11} f'''(C_{a1}^{11})\right)(X_{ab}^1)^2 + 2f''(C_{a1}^{11})X_{ab}^1 \hat{X}_{ab}^1 \right] \right\} \\
&+ \frac{1}{2}\log\det\left( A - \begin{bmatrix} X^0 \\ \hat{X}^0 \\ X^1 \\ \hat{X}^1 \end{bmatrix}^T \begin{bmatrix} C^{00} & iR^{00} & C^{01} & iR^{01} \\ iR^{00} & D^{00} & iR^{10} & D^{01} \\ (C^{01})^T & (iR^{10})^T & C^{11} & iR^{11} \\ (iR^{01})^T & (D^{01})^T & iR^{11} & D^{11} \end{bmatrix}^{-1} \begin{bmatrix} X^0 \\ \hat{X}^0 \\ X^1 \\ \hat{X}^1 \end{bmatrix} \right) .
\end{aligned}
\quad \text{(B.4)}
$$

As usual in these quenched Franz–Parisi style computations, the saddle point expressions for the variables $\mathcal{Q}$ in the joint limits of $m$, $n$, and $\ell$ to zero are independent of $\mathcal{Q}_x$, and so these quantities take the same value they do for the two-point complexity that we computed above. The saddle point conditions for the variables $\mathcal{Q}_x$ are found by extremizing with respect to the action once the variables $\mathcal{Q}$ from the complexity have been fixed.

To evaluate this expression, we need a sensible Ansatz for the variables $\mathcal{Q}_x$. The matrix $A$ we expect to be an ordinary hierarchical matrix, and since the model is a spherical 2-spin the finite but low temperature order will be replica symmetric with nonzero $a_0$. The expected

form of the $X$ matrices follows our reasoning for the 01 matrices of the Appendix A.4: namely, they should have constant rows and a column structure which matches that of the level of RSB order associated with the degrees of freedom that parameterize the columns. Since both the reference configurations and the constrained configurations have replica symmetric order, we expect

$$
X^0 = \begin{bmatrix} x_0 & \cdots & x_0 \\ 0 & \cdots & 0 \\ \vdots & \ddots & \vdots \\ 0 & \cdots & 0 \end{bmatrix} \begin{matrix} \uparrow \\ m-1 \\ \downarrow \end{matrix}, \quad \hat{X}^0 = \begin{bmatrix} \hat{x}_0 & \cdots & \hat{x}_0 \\ 0 & \cdots & 0 \\ \vdots & \ddots & \vdots \\ 0 & \cdots & 0 \end{bmatrix},
$$

$$
X^1 = \begin{bmatrix} 0 & \cdots & 0 \\ x_1 & \cdots & x_1 \\ \vdots & \ddots & \vdots \\ x_1 & \cdots & x_1 \end{bmatrix} \begin{matrix} \uparrow \\ n-1 \\ \downarrow \end{matrix}, \quad \hat{X}^1 = \begin{bmatrix} \hat{x}_1^0 & \cdots & \hat{x}_1^0 \\ \hat{x}_1^1 & \cdots & \hat{x}_1^1 \\ \vdots & \ddots & \vdots \\ \hat{x}_1^1 & \cdots & \hat{x}_1^1 \end{bmatrix}.
$$

(B.5)

with $\leftarrow \ell \rightarrow$ labeling the columns.

Here, the lower blocks of the 0 matrices are zero, because the replicas whose overlap they represent (that of the normalization of the reference configuration) have no correlation with the reference or anything else. The first row of the $X^1$ matrix needs to be zero because of the constraint that the tangent space vectors lie in the tangent plane to the sphere, and therefore have $\mathbf{x}_a \cdot \mathbf{s}_1 = 0$ for any $a$. This produces five parameters to deal with, which we compile in the vector $\mathcal{X} = (x_0, \hat{x}_0, x_1, \hat{x}_1^1, \hat{x}_1^0)$.

Inserting this Ansatz is straightforward in the first part of (B.4), but the term with $\log \det$ is again more complicated. We must invert the block matrix inside. We define

$$
\begin{bmatrix} C^{00} & iR^{00} & C^{01} & iR^{01} \\ iR^{00} & D^{00} & iR^{10} & D^{01} \\ (C^{01})^T & (iR^{10})^T & C^{11} & iR^{11} \\ (iR^{01})^T & (D^{10})^T & iR^{11} & D^{11} \end{bmatrix}^{-1} = \begin{bmatrix} M_{11} & M_{12} \\ M_{12}^T & M_{22} \end{bmatrix},
$$

(B.6)

where the blocks inside the inverse are given by

$$
M_{11} = \left( \begin{bmatrix} C^{00} & iR^{00} \\ iR^{00} & D^{00} \end{bmatrix} - \begin{bmatrix} C^{01} & iR^{01} \\ iR^{10} & D^{01} \end{bmatrix} \begin{bmatrix} C^{11} & iR^{11} \\ iR^{11} & D^{11} \end{bmatrix}^{-1} \begin{bmatrix} C^{01} & iR^{01} \\ iR^{10} & D^{01} \end{bmatrix}^T \right)^{-1},
$$

(B.7)

$$
M_{12} = -M_{11} \begin{bmatrix} C^{01} & iR^{01} \\ iR^{10} & D^{01} \end{bmatrix} \begin{bmatrix} C^{11} & iR^{11} \\ iR^{11} & D^{11} \end{bmatrix}^{-1},
$$

(B.8)

$$
M_{22} = \left( \begin{bmatrix} C^{11} & iR^{11} \\ iR^{11} & D^{11} \end{bmatrix} - \begin{bmatrix} C^{01} & iR^{01} \\ iR^{10} & D^{01} \end{bmatrix}^T \begin{bmatrix} C^{00} & iR^{00} \\ iR^{00} & D^{00} \end{bmatrix}^{-1} \begin{bmatrix} C^{01} & iR^{01} \\ iR^{10} & D^{01} \end{bmatrix} \right)^{-1}.
$$

(B.9)

Here, $M_{22}$ is the inverse of the matrix already analyzed as part of (A.17). Following our discussion of the inverses of block replica matrices above, and reasoning about their products with the rectangular block-constant matrices, things can be worked out using a computer algebra system. For instance, the second term in $M_{11}$ contributes nothing once the appropriate limits are taken, because each contribution is proportional to $n$.

The contribution from the product with the block inverse matrix can be written as

$$
\begin{bmatrix} X_0 \\ i\hat{X}_0 \end{bmatrix}^T M_{11} \begin{bmatrix} X_0 \\ i\hat{X}_0 \end{bmatrix} + 2 \begin{bmatrix} X_0 \\ i\hat{X}_0 \end{bmatrix}^T M_{12} \begin{bmatrix} X_1 \\ i\hat{X}_1 \end{bmatrix} + \begin{bmatrix} X_1 \\ i\hat{X}_1 \end{bmatrix}^T M_{22} \begin{bmatrix} X_1 \\ i\hat{X}_1 \end{bmatrix},
$$

(B.10)

and without too much reasoning one can see that the result is an $\ell \times \ell$ constant matrix. If $A$ is a replica matrix and $c$ is a constant, then

$$\log \det(A - c) = \log \det A - \frac{c}{\sum_{i=0}^{k}(a_{i+1} - a_i)x_{i+1}}, \tag{B.11}$$

where $a_{k+1} = 1$ and $x_{k+1} = 1$. The basic form of the action is therefore (for replica symmetric $A$)

$$2\mathcal{S}_x(\mathcal{Q}_x \mid \mathcal{Q}) = -\beta\mu_1 + \frac{1}{2}\beta^2 f''(1)(1 - a_0^2) + \log(1 - a_0) + \frac{a_0}{1 - a_0} + \mathcal{X}^T\left(\beta B - \frac{1}{1 - a_0}C\right)\mathcal{X}, \tag{B.12}$$

where the matrix $B$ comes from the $\mathcal{X}$-dependent parts of the first lines of (B.4) and is given by

$$B = \begin{bmatrix} \hat{\beta}_0 f''(q) + r_{10}f'''(q) & f''(q) & 0 & 0 & 0 \\ f''(q) & 0 & 0 & 0 & 0 \\ 0 & 0 & -\hat{\beta}_1 f''(q_0^{11}) - r_0^{11}f'''(q_0^{11}) & -f''(q_0^{11}) & 0 \\ 0 & 0 & -f''(q_0^{11}) & 0 & 0 \\ 0 & 0 & 0 & 0 & 0 \end{bmatrix}, \tag{B.13}$$

and where the matrix $C$ encodes the coefficients of the quadratic form (B.10), and is given element-wise by

$$C_{11} = d_d^{00}f'(1), \qquad C_{12} = r_d^{00}f'(1), \qquad C_{22} = -f'(1),$$

$$C_{13} = \frac{1}{1 - q_0}\left((r_d^{11} - r_0^{11})\left(r^{01} - q\frac{r_d^{11} - r_0^{11}}{1 - q_0}\right)(f'(1) - f'(q_0)) + qf'(1)d_d^{00}\right.$$

$$\left. + r_d^{00}(r^{10}f'(1) + (r_d^{11} - r_0^{11})f'(q))\right),$$

$$C_{15} = r_d^{00}f'(q) + \left(r^{01} - q\frac{r_d^{11} - r_0^{11}}{1 - q_0}\right)(f'(1) - f'(q_0)), \qquad C_{14} = -C_{15},$$

$$C_{23} = \frac{1}{1 - q_0}\left((qr_d^{00} - r^{10})f'(1) - (r_d^{11} - r_0^{11})f'(q)\right), \qquad C_{24} = f'(q), \qquad C_{25} = -C_{24}, \tag{B.14}$$

$$C_{33} = -\frac{r_d^{11} - r_0^{11}}{1 - q_0}\left[\frac{r_d^{11} - r_0^{11}}{1 - q_0}f'(1) - 2\left(\frac{qr^{01} - r_0^{11}}{1 - q_0} + \frac{1 - q^2}{1 - q_0}\frac{r_d^{11} - r_0^{11}}{1 - q_0}\right)(f'(1) - f'(q_0))\right.$$

$$\left. - 2\frac{qr^{00} - r^{10}}{1 - q_0}f'(q)\right] - \frac{1 - q^2}{(1 - q_0)^2}\frac{(r^{10} - qr_d^{00})^2}{(1 - q_0)^2}f'(1),$$

$$C_{34} = -(qr^{01} - r_0^{11})\frac{f'(1) - f'(q_0)}{1 - q_0} - \frac{r_d^{11} - r_0^{11}}{1 - q_0}\left(\frac{1 - q^2}{1 - q_0}(f'(1) - f'(q_0)) - f'(q_0)\right)$$

$$- f'(q)\frac{qr_d^{00} - r^{10}}{1 - q_0},$$

$$C_{35} = -C_{34} - \frac{r_d^{11} - r_0^{11}}{1 - q_0}(f'(1) - f'(q_0)), \qquad C_{44} = f'(1) - 2f'(q_0), \qquad C_{45} = f'(q_0),$$

$$C_{55} = -f'(1).$$

The saddle point conditions read

$$0 = -\beta^2 f''(1)a_0 + \frac{a_0 - \mathcal{X}^T C \mathcal{X}}{(1 - a_0)^2}, \qquad\qquad 0 = \left(\beta B - \frac{1}{1 - a_0}C\right)\mathcal{X}. \tag{B.15}$$

Note that the second of these conditions implies that the quadratic form in $\mathcal{X}$ in the action vanishes at the saddle.

We would like to take the limit of $\beta \to \infty$. As is usual in the two-spin model, the appropriate limit of the order parameter is $a_0 = 1 - (y\beta)^{-1}$. Upon inserting this scaling and taking the limit, we finally find

$$\lambda_{\min} = -2 \lim_{\beta \to \infty} \frac{1}{\beta} \mathcal{S}_x = \mu_1 - \left( y + \frac{1}{y} f''(1) \right),$$ (B.16)

with associated saddle point conditions

$$0 = -f''(1) + y^2(1 - \mathcal{X}^T C \mathcal{X}), \qquad 0 = (B - yC)\mathcal{X},$$ (B.17)

as reported in the main text.

The solution described here also encodes information about the correlation between the eigenvector $\mathbf{x}_{\min}$ associated with the minimum eigenvalue and the tangent direction connecting the two stationary points $\mathbf{x}_{0 \leftarrow 1}$. The overlap between these vectors is directly related to the value of the order parameter $x_0 = \frac{1}{N} \boldsymbol{\sigma}_1 \cdot \mathbf{x}_a$. This tangent vector is $\mathbf{x}_{0 \leftarrow 1} = \frac{1}{1-q}(\boldsymbol{\sigma}_1 - q \mathbf{s}_a)$, which is normalized and lies strictly in the tangent plane of $\mathbf{s}_a$. Then the overlap between the two vectors is

$$q_{\min} = \frac{\mathbf{x}_{0 \leftarrow 1} \cdot \mathbf{x}_{\min}}{N} = \frac{x_0}{1-q},$$ (B.18)

where $\mathbf{x}_{\min} \cdot \mathbf{s}_a = 0$ because of the restriction of the $\mathbf{x}$ vectors to the tangent plane at $\mathbf{s}_a$.

# C   Comparison with the Franz–Parisi potential

The comparison between the Franz–Parisi potential at zero temperature and the minimum-energy limit of the two-point complexity is of interest to some specialists because the two computations qualitatively describe the same thing. However, it was previously found that the two computations produce different results in the pure spherical models, to the surprise of those researchers [13]. Understanding this difference is subtle. The zero-temperature Franz–Parisi potential underestimates the energy where nearby minima are found, because it includes any configuration that is a minimum on the subspace created by constraining the overlap. Many of these configurations will not have zero gradient perpendicular to the overlap constraint manifold, and therefore are not proper minima of the energy.

A strange feature of the comparison for the pure spherical models was that the two-point complexity and the Franz–Parisi potential coincided at their local maximum in $q$. It is not clear why this coincidence occurs, but it is good news for those who use the Franz–Parisi potential to estimate the height of the free energy barrier between states. Though it everywhere else underestimates the energy of nearby states, it correctly gives the value of this highest barrier.

Here, we compute the Franz–Parisi potential for the mixed spherical models at zero temperature, with respect to a reference configuration fixed to be a stationary point of energy $E_0$ and stability $\mu_0$ [32, 33]. Comparing with the lower energy boundary of the 2-point complexity, we find that the story in the mixed models is the same as that in the pure models: the Franz–Parisi potential underestimates the lowest energy of nearby minima almost everywhere except at its peak, where the two measures coincide.

The potential is defined as the average free energy of a system constrained to lie with a fixed overlap $q$ with a reference configuration (here a stationary point with fixed energy and stability), and given by

$$\beta V_\beta(q \mid E_0, \mu_0) = -\frac{1}{N} \overline{\int \frac{d\nu_H(\boldsymbol{\sigma}, \varsigma \mid E_0, \mu_0)}{\int d\nu_H(\boldsymbol{\sigma}', \varsigma' \mid E_0, \mu_0)} \log\left( \int d\mathbf{s} \, \delta\big(\|\mathbf{s}\|^2 - N\big) \delta(\boldsymbol{\sigma} \cdot \mathbf{s} - Nq) e^{-\beta H(\mathbf{s})} \right)}.$$
(C.1)

Both the denominator and the logarithm are treated using the replica trick, which yields

$$
\begin{aligned}
&\beta V_\beta(q \mid E_0, \mu_0) \\
&= -\frac{1}{N} \lim_{\substack{m \to 0 \\ n \to 0}} \frac{\partial}{\partial n} \overline{\int \left( \prod_{b=1}^m d\nu_H(\boldsymbol{\sigma}_b, \varsigma_b \mid E_0, \mu_0) \right) \left( \prod_{a=1}^n d\mathbf{s}_a\, \delta(\|\mathbf{s}_a\|^2 - N)\, \delta(\boldsymbol{\sigma}_1 \cdot \mathbf{s}_a - Nq)\, e^{-\beta H(\mathbf{s}_a)} \right)}.
\end{aligned}
\tag{C.2}
$$

The derivation of this proceeds in much the same way as for the complexity or the isolated eigenvalue. Once the $\delta$-functions are converted to exponentials, the $H$-dependent terms can be expressed by convolution with the linear operator

$$
\mathcal{O}(\mathbf{t}) = \sum_a^m \delta(\mathbf{t} - \boldsymbol{\sigma}_a)\left(i\hat{\boldsymbol{\sigma}}_a \cdot \partial_{\mathbf{t}} - \hat{\beta}_0\right) - \beta \sum_a^n \delta(\mathbf{t} - \mathbf{s}_a).
\tag{C.3}
$$

Averaging over $H$ squares the application of this operator to $f$ as before. After performing a Hubbard–Stratonovich using matrix order parameters identical to those used in the calculation of the complexity, we find that

$$
\beta V_\beta(q \mid E_0, \mu_0) = -\frac{1}{N} \lim_{\substack{m \to 0 \\ n \to 0}} \frac{\partial}{\partial n} \int d\mathcal{Q}_0\, d\mathcal{Q}_1\, e^{Nm\mathcal{S}_0(\mathcal{Q}_0) + Nn\mathcal{S}_{\mathrm{FP}}(\mathcal{Q}_1 \mid \mathcal{Q}_0)},
\tag{C.4}
$$

where $\mathcal{S}_0$ is the same as in (A.16) and

$$
\begin{aligned}
n\mathcal{S}_{\mathrm{FP}} = {}&\frac{1}{2}\beta^2 \sum_{ab}^n f(Q_{ab}) + \beta \sum_a^m \sum_b^n \left[ \hat{\beta}_0 f(C_{ab}^{01}) + R_{ab}^{10} f'(C_{ab}^{01}) \right] \\
&+ \frac{1}{2} \log \det \left( Q - \begin{bmatrix} C^{01} \\ iR^{10} \end{bmatrix}^T \begin{bmatrix} C^{00} & iR^{00} \\ iR^{00} & D^{00} \end{bmatrix}^{-1} \begin{bmatrix} C^{01} \\ iR^{10} \end{bmatrix} \right).
\end{aligned}
\tag{C.5}
$$

Here, because we are at low but nonzero temperature for the constrained configuration, we make a 1RSB anstaz for the matrix $Q$, while the 00 matrices will take their saddle point value for the one-point complexity and the 01 matrices have the same structure as (A.22). Inserting these gives

$$
\begin{aligned}
\beta V_\beta = {}&\frac{1}{2}\beta^2\left[ f(1) - (1-x)f(q_1) - xf(q_0) \right] + \beta\hat{\beta}_0 f(q) + \beta r^{10} f'(q) - \frac{1-x}{x} \log(1 - q_1) \\
&+ \frac{1}{x} \log(1 - (1-x)q_1 - xq_0) + \frac{q_0 - d_d^{00} f'(1)q^2 - 2r_d^{00} f'(1)r^{10}q + (r^{10})^2 f'(1)}{1 - (1-x)q_1 - xq_0}.
\end{aligned}
\tag{C.6}
$$

The saddle point for $r^{10}$ can be taken explicitly. After this, we take the limit of $\beta \to \infty$. There are two possibilities. First, in the replica symmetric case $x = 1$, and in the limit of large $\beta$ $q_0$ will scale like $q_0 = 1 - (y_0\beta)^{-1}$. Inserting this, the limit is

$$
V_\infty^{\mathrm{RS}} = -\hat{\beta}_0 f(q) - r_d^{00} f'(q)q - \frac{1}{2}\left( y_0(1 - q^2) + \frac{f'(1)^2 - f'(q)^2}{y_0 f'(1)} \right).
\tag{C.7}
$$

The saddle point in $y_0$ can now be taken, taking care to choose the solution for $y_0 > 0$. This gives

$$
V_\infty^{\mathrm{RS}}(q \mid E_0, \mu_0) = -\hat{\beta}_0 f(q) - r_d^{00} f'(q)q - \sqrt{(1 - q^2)\left( 1 - \frac{f'(q)^2}{f'(1)^2} \right)}.
\tag{C.8}
$$

The second case is when the inner statistical mechanics problem has replica symmetry breaking. Here, $q_0$ approaches a nontrivial limit, but $x = z\beta^{-1}$ approaches zero and $q_1 = 1 - (y_1\beta)^{-1}$

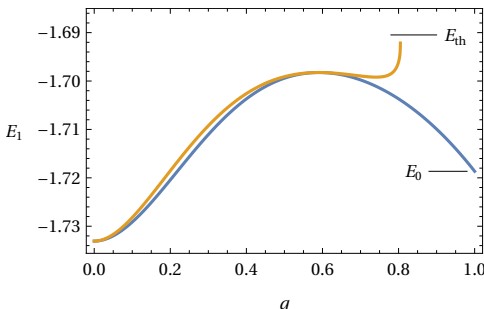

Figure 10: Comparison of the lowest-energy stationary points at overlap $q$ with a reference minimum of $E_0 = -1.71865 < E_{\mathrm{th}}$ and $\mu_0 = 6.1 > \mu_{\mathrm{m}}$ (yellow, top), and the zero-temperature Franz–Parisi potential with respect to the same reference minimum (blue, bottom). The two curves coincide precisely at their minimum $q = 0$ and at the local maximum $q \simeq 0.5909$.

approaches one. The result is

$$
\begin{aligned}
V_{\infty}^{1\mathrm{RSB}}(q \mid E_0, \mu_0) = {} & -\hat{\beta}_0 f(q) - r_{\mathrm{d}}^{00} f'(q) q - \frac{1}{2}\bigg( z(f(1) - f(q_0)) + \frac{f'(1)}{y_1} - \frac{y_1(q^2 - q_0)}{1 + y_1 z(1 - q_0)} \\
& - (1 + y_1 z(1 - q_0))\frac{f'(q)^2}{y_1 f'(1)} + \frac{1}{z}\log\left(1 + z y_1(1 - q_0)\right) \bigg).
\end{aligned}
$$
(C.9)

Though the saddle point in $y_1$ can be evaluated in this expression, it delivers no insight. The final potential is found by taking the saddle over $z$, $y_1$, and $q_0$. A plot comparing the result to the minimum energy saddles is found in Fig. 10. As noted above, there is little qualitatively different from what was found in [13] for the pure models.

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
