# Peer review of "Arrangement of nearby minima and saddles in the mixed spherical energy landscapes"

_SciPost Physics, doi:SciPost Phys. 16, 001 (2024)_

## Round 2 · Referee Report · Anonymous (Referee 1) · 2023-10-10

Strengths

1- In-depth replica-symmetric computation of the complexity of the mixed p-spin model, with special emphasis on the 3+4 model analysed by Folena et al. 2- Very well written with summary sections throughout.

Weaknesses

1- At times, the paper becomes overly technical. Some of the computations could be moved to the appendix to promote a more fluid and physically oriented reading. 2- Local stability analysis seems not to be enough to characterise and suitably distinguish the dynamical attractors.

Report

I believe that the paper meets the requirements of the journal and deserves to be published. I have included a list of suggested minor changes though.

i) On page 7, when referring to the set of marginal states that attract dynamics "as evidenced by power-law relaxations", it would be convenient to provide references for this statement.

ii) On the same page, the author refers to a quadratic pseudo-gap in the complexity function associated with marginal states. It would be helpful to have some more indication of how this was derived or, again, to provide appropriate references.

iii) Section 4 “Calculation of the two-point complexity”. The author states that conditioning the Hessian matrix of the stationary points to have a given energy and given stability properties influences the statistics of points only at the sub-leading order. It would be valuable to clarify the conditions under which this occurs. I was thus wondering whether the author can straightforwardly generalize such a computation and give some insights in the case of a sparse (no longer fully connected) model.

iv) Eq. (34) is quite complicated and difficult to grasp by eye. I thus wonder whether the numerical protocol is robust enough to be sure that by initializing differently, not exactly at q=0, the same solution is always found. How sensitive is the protocol to the choice of initial conditions?

v) In Section 5, the analysis of an isolated eigenvalue, which can be attributed to a low-rank perturbation in the Hessian matrix, is discussed. The technique results from a generalization of a paper recently published by H. Ikeda, restricted to a quadratic model though. It would be worthwhile to discuss how many of these predictions can be extended to models defined by a double-well potential or to optimization problems relying on non-quadratic functions (such as ReLu, sigmoid).

Requested changes

I found the paper interesting but quite technical in some points. Moving the saddle-point computations and part of the analysis (see for instance on pages 11-13 and 18-20) to the supplement would make it easier to capture the main results, especially for general readers without extensive expertise in the replica trick and these models.

  • validity: high
  • significance: high
  • originality: good
  • clarity: high
  • formatting: excellent
  • grammar: perfect

Author:  Jaron Kent-Dobias  on 2023-12-05  [id 4169]

(in reply to Report 1 on 2023-10-10)
Category:
answer to question

We thank the referee for their positive assessment. We believe their concerns with the manuscript have been addressed in the updated version. Here we address their specific concerns.

i) On page 7, when referring to the set of marginal states that attract dynamics "as evidenced by power-law relaxations", it would be convenient to provide references for this statement.

The evidence of power-law relaxation to marginal minima is contained in G. Folena and F. Zamponi, On weak ergodicity breaking in mean-field spin glasses, SciPost Physics 15(3), 109 (2023). In the original manuscript this work was cited at the end of the sentence, but the sentence has now be rephrased and the specific point about power-law relaxation has been removed to improve clarity.

ii) On the same page, the author refers to a quadratic pseudo-gap in the complexity function associated with marginal states. It would be helpful to have some more indication of how this was derived or, again, to provide appropriate references.

The form of the pseudo-gap in overlap for marginal states above the threshold energy is demonstrated in the subsection on the expansion of the complexity in the near neighborhood (equation (40) in the original manuscript, equation (17) in the revised manuscript). In the revised manuscript more has been done to emphasize the of the pseudogap analysis.

iii) Section 4 “Calculation of the two-point complexity”. The author states that conditioning the Hessian matrix of the stationary points to have a given energy and given stability properties influences the statistics of points only at the sub-leading order. It would be valuable to clarify the conditions under which this occurs. I was thus wondering whether the author can straightforwardly generalize such a computation and give some insights in the case of a sparse (no longer fully connected) model.

In the manuscript, we have added a small clarification as to the reason for this:

"This is because the conditioning amounts to a rank-one perturbation to the Hessian matrix, which does not affect the bulk of its spectrum."

From this, one can reason that the same assumptions will hold whenever rank-one perturbations do not affect the bulk spectrum. While we are not experts in the theory of sparse matrices, it seems likely this condition breaks down when a matrix is sufficiently sparse.

iv) Eq. (34) is quite complicated and difficult to grasp by eye. I thus wonder whether the numerical protocol is robust enough to be sure that by initializing differently, not exactly at q=0, the same solution is always found. How sensitive is the protocol to the choice of initial conditions?

It is true that equation (34) (now (11)) is quite complicated, but the numeric methods we use find the same solutions quite robustly. First, we make use of arbitrary precision arithmetic in Mathematica to ensure that the roots of the saddle point equations derived from this expression are indeed good roots, in this case with a 30-digit working precision. Second, the initialization near a good solution known analytically, namely the solution at q = 0, is crucial because if initialized from random conditions a valid solution is never found. If we initialize the root-finding algorithm using the known solution at q = 0 and then attempt to solve the equations at some small q > 0, a consistent solution is found so long as q is sufficiently small. This is also true if the initial condition is randomly perturbed by a small amount. If the first q > 0 is too large or the random perturbation is too large, only nonphysical solutions are found. Luckily, we expect that the complexity of stationary points at different proximities varies smoothly, so that this procedure is justified.

v) In Section 5, the analysis of an isolated eigenvalue, which can be attributed to a low-rank perturbation in the Hessian matrix, is discussed. The technique results from a generalization of a paper recently published by H. Ikeda, restricted to a quadratic model though. It would be worthwhile to discuss how many of these predictions can be extended to models defined by a double-well potential or to optimization problems relying on non-quadratic functions (such as ReLu, sigmoid).

The technique is quite generic, and the ability to apply it to other models rests mostly in the tractability of the saddle point calculation. I guess the reviewer is referencing the KHGPS model, or simple neural networks. The principle challenge in these cases is the Kac–Rice calculation itself, which has not been quantitatively extended to systems with non-Gaussian disorder. If this were resolved, using this technique to analyze the properties of an isolated eigenvalue would be a painful corollary. (ReLu is problematic with respect to these landscape methods, however, because it does not have a well-defined Hessian everywhere.)

---

## Round 2 · Referee Report · Anonymous (Referee 2) · 2023-11-19

Strengths

1- The work proposes a novel analysis of the landscape of the mixed p-spin model using a two-point Kac-Rice formula. 2- The equations are used to study marginal minima, showing that they are always isolated except at the threshold energy. 3- In general, the paper provides several insights into the landscape of the model. Despite it was not possible to connect these properties to dynamics yet, they may provide valuable insights for future investigations.

Weaknesses

1- Clarity.

Report

This paper presents several insights into an open problem. Despite the new insights did not allow the author to connect dynamics and landscape properties, they represent a significant contribution that I believe is worth publishing. Except for a few conceptual questions that I hope the author clarifies in a rebuttal, my main concern is clarity. At the moment the paper looks unpolished and hard to read.

Comments/Questions

On page 9, the author points out that there are solutions with complexity 0 that do not show an extensive barrier "in any situation". First, this "in any situation" is quite unclear. Does the author mean above and below the threshold energy? Does this solution exist even at high energy? Can the author comment on what this solution could imply?

At the technical level, I am confused by one of the constraints imposed in eq.16, when \sigma_1 couples with all replicated s_a. I was expecting a sum of \sigma_b.s_a over a and b. This may represent a rotation applied to all replicas along a reference direction, which is probably what the author did, but there is no comment about that. In general, it would have helped to specify the constraints enforced instead of writing simply "Lagrange multipliers" before eq.16.

The author analyses the problem using the Franz-Parisi potential, however, this analysis does not seem to matter in the paper. We can read a comment at the end of Sec.3.1 but without actual implications. It should be either removed or expanded, at the moment it seems just without purpose.

In general, the paper lacks in clarity. It has several sentences that miss a conclusion, paragraphs that seem out of context, references missing, and the final sections unpolished. I will proceed in order

  1. Introduction: • "We see arrangements of barriers relative to each other, perhaps...". Why "perhaps"? Second, where is this analysis carried out? In the results section, the author analyses stable minima and marginal states, I don't know where to look. Adding a reference would have helped.
  2. Model • After eq.3, the author comments on the replica ansatz, but this is out of place. We are still introducing the model. It would be better to have it at the end of the section (where indeed the author comes back to the same concept) or remove it entirely. • fig.1, add a caption under each figure saying what they are (oriented saddles, oriented minima, etc), it is much easier to read. • fig.2, elaborate a bit more in the main text. This is introduced at the of the section without any comment.
  3. Results • fig.3, "the dot-dashed lines on both plots depict the trajectory of the solid line on the other plot", which one? • fig.3, "In this case, the points lying nearest to the reference minimum are saddle with mu<mu, but with energies smaller than the threshold energy", so? What is the implication? This misses a conclusion. • Sec.3.1, the author comments on the similarity with the pure model, without explaining what is similar. What should we expect on the p-spin? At least the relevant aspects. It would also be useful to plot a version of Fig.3 for the p-spin. It would make the discussion easier to follow. • "the nearest neighbour points are always oriented saddles", where do I see this? • the sentence "like in the pure models, the emergence [...]" is extremely hard to parse and the paragraph ends without a conclusion. What are the consequences? • at page 9, the author talk about \Sigma_12 that however has not been defined yet.
  4. Calculation • this section starts without explaining what is the strategy to solve the problem. Explaining how the following subsection will contribute to the solution without entering into the details of the computation would be of great help. • "This replica symmetry will be important later" how? Either we have an explanation following or it should be removed. • at the end of a step it would be good to wrap everything up. For instance, sec.4.2 ends with "we do not include these details, which are standard" at least give a reference. Second, add the final result. • "there is a desert where none are found" -> solutions are exponentially rare (or something else)

I would suggest a rewriting, especially the last sessions (4-6). I understand the intention of removing simple details, but they should be replaced by comments. The impression (which can be wrong but gives the idea) is of some working notes where simple steps have been removed, resulting in hard-to-follow computations. Finally, I would also recommend moving these sections to an appendix (after acknowledgement and funding).

  • validity: high
  • significance: good
  • originality: high
  • clarity: poor
  • formatting: reasonable
  • grammar: acceptable

Author:  Jaron Kent-Dobias  on 2023-12-05  [id 4170]

(in reply to Report 2 on 2023-11-19)
Category:
answer to question

We thank the referee for their positive assessment of our scientific work, and their many pieces of constructive feedback. The revised manuscript has substantially improved as a result. Here, we address their specific concerns.

On page 9, the author points out that there are solutions with complexity 0 that do not show an extensive barrier "in any situation". First, this "in any situation" is quite unclear. Does the author mean above and below the threshold energy? Does this solution exist even at high energy? Can the author comment on what this solution could imply?

Here "any situation" means "for a reference point of any energy and stability." This includes energies above and below the threshold energy, at stabilities that imply saddles, minima, or marginal minima, and even for combinations of energy and stability where the complexity of stationary points is negative.

The two-point complexity is computed under the condition that the reference point exists. Given that the reference point exists, there is at least one point that can be found at zero overlap with the reference: itself. This reasoning alone rationalizes why we should find a solution with Σ₁₂ = 0, q = 0, and E₁ = E₀, μ₁ = μ₀ for any E₀ and μ₀.

The interpretation is more subtle. Two different stationary points cannot lie at the same point, but the complexity calculation only resolves numbers of points that are exponential in N and differences in overlap that are linear in N. Therefore, the complexity calculation is compatible with many stationary points being contained in the subextensive region of dimension Δq = O(1/N) around any reference point. We can reason as to where these extremely near neighbors are likely or unlikely to exist in specific conditions, but the complexity calculation cannot rule them out.

This is point is not crucial to anything in the paper, except to make more precise the statement that non-threshold marginal minima are separated by a gap in their overlap. Because marginal minima have very flat directions, they are good candidates for possessing these extremely near neighbors, and this might lead one to say they are not isolated. However, if such extremely near neighbors exist, they are irrelevant to dynamics: the entire group is isolated, since the complexity of similar stationary points at a small but extensive overlap further is negative.

Because the point is not important to the conclusions of the paper, the paragraph has been revised for clarity and moved to a footnote.

At the technical level, I am confused by one of the constraints imposed in eq.16, when \sigma_1 couples with all replicated s_a. I was expecting a sum of \sigma_b.s_a over a and b. This may represent a rotation applied to all replicas along a reference direction, which is probably what the author did, but there is no comment about that. In general, it would have helped to specify the constraints enforced instead of writing simply "Lagrange multipliers" before eq.16.

The fact noticed by the referee that only σ₁ appears in the scalar product with sₐ in equation (16) of the original manuscript (now equation (33)) was not introduced in that equation, but instead was introduced in equation (10). Right after that equation, the special status of σ₁ was clarified. This arises because of the structure of equation (9): in that equation, the logarithmic expression being averaged depends only on σ, which corresponds with σ₁ in the following equation. σ₂ through σₘ correspond to σ', which is replicated (m - 1) times to bring the normalization into the numerator. Therefore, there is a clear reason behind the asymmetry among the replicas associated with the reference spin, and it was not due to an ad-hoc transformation as suggested by the referee.

As part of the rewriting of the manuscript for clarity, this subtlety has been emphasized around equations (9) and (10). In the revised manuscript, the specific comment regarding Lagrange multipliers is no longer present, but they referred to the variables ω introduced in the Model section.

The author analyses the problem using the Franz-Parisi potential, however, this analysis does not seem to matter in the paper. We can read a comment at the end of Sec.3.1 but without actual implications. It should be either removed or expanded, at the moment it seems just without purpose.

The analysis of the Franz-Parisi potential has been moved to Appendix C, with a more explanatory discussion of our interest in it included in the manuscript. In short, the referee is right to point out that it has no implications for the main topic of the paper. It is included because some specialists will be interested in the comparison between it and the two-point complexity. This reasoning is now explained at the beginning of Appendix C.

"We see arrangements of barriers relative to each other, perhaps...". Why "perhaps"? Second, where is this analysis carried out? In the results section, the author analyses stable minima and marginal states, I don't know where to look. Adding a reference would have helped.

The sentence in question has now been rephrased, but "perhaps" was due to the fact that not very much is learned about the mutual arrangement of saddles from this work. In order to make clear what conclusions can be drawn about saddles from our calculation, we have added a new subsection to the Results section, 3.2: Grouping of saddle points. This subsection contains two paragraphs detailing what one might want to know about the geometry of saddle points, and what we actually learn from the two-point complexity.

After eq.3, the author comments on the replica ansatz, but this is out of place. We are still introducing the model. It would be better to have it at the end of the section (where indeed the author comes back to the same concept) or remove it entirely.

The referee is right to point out this oversight, and the note about the specific influence of the covariance function f on the form of RSB has been moved into the details for the calculation of the complexity, in subsection A.4: Replica ansatz and saddle point. Where it was in section 2 we now say

"The choice of f has significant effect on the form of equilibrium order in the model, and likewise influences the geometry of stationary points."

fig.1, add a caption under each figure saying what they are (oriented saddles, oriented minima, etc), it is much easier to read.

The suggestion of the referee was good and was implemented in the new manuscript.

fig.2, elaborate a bit more in the main text. This is introduced at the of the section without any comment.

A paragraph discussing Fig. 2 has been added to the main text, and the end of Section 2.

fig.3, "the dot-dashed lines on both plots depict the trajectory of the solid line on the other plot", which one?

The answer is both. This confusing sentence has been clarified in the new manuscript. It now reads:

"The dot-dashed line on the left plot depicts the trajectory of the solid line on the right plot, and the dot-dashed line on the right plot depicts the trajectory of the solid line on the left plot."

fig.3, "In this case, the points lying nearest to the reference minimum are saddle with mu\<mu, but with energies smaller than the threshold energy", so? What is the implication? This misses a conclusion.

These low-lying saddles represent large deviations from the typical complexity. The point has been clarified by appending "which makes them an atypical population of saddles" to the sentence.

Sec.3.1, the author comments on the similarity with the pure model, without explaining what is similar. What should we expect on the p-spin? At least the relevant aspects. It would also be useful to plot a version of Fig.3 for the p-spin. It would make the discussion easier to follow.

In the reversed manuscript, the points of comparison with the pure models are made more explicit, as the referee suggests. We do not think it is necessary to include a figure for the pure models, instead clarifying the most important departure in the text:

"The largest difference between the pure and mixed models is the decoupling of nearby stable points from nearby low-energy points: in the pure p-spin model, the left and right panels of Fig. 3 would be identical up to a constant factor −p."

For those interested in more detailed comparisons, the relevant figure for the pure models is found in the paper twice cited in that subsection.

"the nearest neighbour points are always oriented saddles", where do I see this?

We have added a sentence to clarify this point:

"This is a result of the persistent presence of a negative isolated eigenvalue in the spectrum of the nearest neighbors, e.g., as in the shaded regions of Fig. 3."

the sentence "like in the pure models, the emergence [...]" is extremely hard to parse and the paragraph ends without a conclusion. What are the consequences?

This sentence has been expanded to make it more clear, and the statement now reads

"Like in the pure models, the minimum energy and maximum stability of nearby points are not monotonic in q: there is a range of overlap where the minimum energy of neighbors decreases with overlap. The transition from stable minima to index-one saddles along the line of lowest-energy states occurs at its local minimum, another similarity with the pure models [13]. This point is interesting because it describes the properties of the nearest stable minima to the reference point. It is not clear why the local minimum of the boundary coincides with this point or what implications that has for behavior."

We also now emphasize that the implications are not known. However, the coincidence itself it interesting, at the very least for the ability to predict where an isolated eigenvalue should destabilize nearby minima without making the computation for the eigenvalue.

at page 9, the author talk about \Sigma_12 that however has not been defined yet.

The referee is correct to point out this oversight, which has now been amended by a qualitative definition of Σ₁₂ at the beginning of the results section.

this section starts without explaining what is the strategy to solve the problem. Explaining how the following subsection will contribute to the solution without entering into the details of the computation would be of great help.

The explanation of the calculation for the complexity has been reorganized and expanded in the new manuscript. In part of this expansion, we added more explanation of this kind. Most of this is now found in Appendix A.

"This replica symmetry will be important later" how? Either we have an explanation following or it should be removed.

The comment has been removed in the new manuscript.

at the end of a step it would be good to wrap everything up. For instance, sec.4.2 ends with "we do not include these details, which are standard" at least give a reference. Second, add the final result.

In the revised manuscript, more has been done to wrap up each section. For instance, what was section 4.2 and is is now section A.2 now ends

"The result of this calculation is found in the effective action (44), where it contributes all terms besides the functions D contributed by the Hessian terms in the previous section and the logarithms contributed by the Hubbard–Stratonovich transformation of the next section."

"there is a desert where none are found" -> solutions are exponentially rare (or something else)

The statement has been rewritten, and now says

"Therefore, marginal minima whose energy E₀ is greater than the threshold have neighbors at arbitrarily close distance with a quadratic pseudogap, while those whose energy is less than the threshold have an overlap gap."

---

## Round 3 · Author Response

Both referees of our first submission gave reports that were positive about the scientific content of our manuscript, but critical of its presentation and organization. Following their advice, we have rewritten large pieces of the manuscript and reorganized it, moving many details of the calculations into appendices and adding more explanation to certain statements and steps that were previously confusing.

---

## Round 3 · List of Changes

It is difficult to list each specific change because of the nature of the suggested amendments. Here is a point-by-point summary of what we did:
- Large portions of the calculations for the two-point complexity, the isolated eigenvalue, and the entirety of the Franz–Parisi potential were moved into appendices.
- The explanation of steps in these calculations was expanded, especially in the calculation of the complexity.
- Much of the text was edited for clarity, with confusing statements amended or removed (including but not limited to those flagged by the referees).
- A new subsection 3.2 was added to the results section briefly detailing the mutual geometry of saddle points implied by the two-point complexity.
- More motivation has been given for our interest in specific results, including several expectations of what might be learned from the two-point complexity that were not borne out.
- More references were added where appropriate.

---

## Editorial Decision

published